# Entropy-Aware On-Policy Distillation of Language Models

Woogyeol Jin [1]   Taywon Min [1]   Yongjin Yang [2 3]   Dennis Wei [4]   Yi Zhou [4]   Swanand Ravindra Kadhe [4]
Nathalie Baracaldo [4]   Kimin Lee [1]

## Abstract

On-policy distillation is a promising approach for transferring knowledge between language models, where a student learns from dense token-level signals along its own trajectories. The standard objective is reverse KL divergence, which encourages the student to match the teacher's high-confidence predictions. However, we show that the mode-seeking property of reverse KL reduces generation diversity and yields unstable learning signals when the teacher distribution has high entropy. To address this, we introduce Entropy-Aware On-Policy Distillation (EOPD), which augments the reverse KL objective with forward KL on tokens where the teacher distribution has high entropy. This captures the full range of plausible outputs at uncertain steps while retaining precise imitation elsewhere, balancing mode-seeking precision with mode-covering robustness without sacrificing on-policy training efficiency. Experiments show that our method maintains generation diversity (sustained token-level entropy) and improves student–teacher alignment (lower forward KL on high-entropy tokens). Across six math reasoning benchmarks, this yields Pass@8 accuracy gains of $+1.37$ for Qwen3-0.6B-Base, $+2.39$ for Qwen3-1.7B-Base, and $+5.05$ for Qwen3-4B-Base compared to baseline on-policy distillation methods. These results demonstrate that accounting for teacher uncertainty is essential for maintaining diversity and achieving effective knowledge transfer. Our code is publicly available at https://github.com/WLS04/EOPD.

## 1. Introduction

Knowledge distillation (Hinton et al., 2015) is a promising approach for transferring the capabilities of large language models (LLMs) to smaller, more efficient models with lower inference cost and improved deployability. Traditional distillation relies on off-policy teacher data, training students with supervised loss or forward KL divergence (Kim & Rush, 2016). However, this introduces a distribution mismatch between training sequences and those generated by the student at inference time.

On-policy distillation addresses this by having the student generate samples that are corrected by the teacher, typically via reverse KL divergence (Gu et al., 2024; Agarwal et al., 2024; Lu & Lab, 2025). As noted by Lu & Lab (2025), this objective can be seamlessly integrated into standard reinforcement learning (RL) pipelines. Furthermore, the approach is highly efficient: recent work (Lu & Lab, 2025; Yang et al., 2025) demonstrates that on-policy distillation can match RL-trained models on math reasoning benchmarks at $10\times$ lower compute cost than GRPO (Shao et al., 2024), which has made on-policy distillation a standard recipe for distillation.

However, reverse KL is a *mode-seeking* objective: while it effectively captures the teacher's dominant modes, we find that it reduces student diversity and yields unstable learning signals at positions where the teacher distribution has high entropy. This limits the student's ability to preserve the teacher's distributional structure when probability mass is spread across multiple tokens. It is particularly problematic in reasoning tasks, where high-entropy tokens often represent key decision points with multiple valid paths (Wang et al., 2025; Cheng et al., 2025). The student then converges to a narrower behavioral repertoire than the teacher, even when reverse KL itself is well optimized.

**Contribution.** To address this limitation, we propose Entropy-Aware On-Policy Distillation (EOPD), a distillation framework that balances efficiency and diversity. Our key insight is that reverse KL and forward KL are complementary: reverse KL enables efficient learning on confident predictions, while forward KL's mode-covering property transfers uncertainty and global structure. Our specific contributions are as follows:

[1]KAIST AI, Seoul, South Korea [2]University of Toronto, Ontario, Canada [3]Vector Institute, Ontario, Canada [4]IBM Research, San Jose, CA, USA. Correspondence to: Kimin Lee <kiminlee@kaist.ac.kr>.

*Proceedings of the $43^{rd}$ International Conference on Machine Learning*, Seoul, South Korea. PMLR 306, 2026. Copyright 2026 by the author(s).

- **Analysis of Diversity Degradation and Training Instability.** We conduct a systematic analysis of token-level entropy (§3.1), revealing that standard on-policy distillation causes diversity collapse, retaining only 6.8% of high-entropy tokens compared to 18.5% in the teacher. Furthermore, through a controlled toy experiment (§3.2), we demonstrate that the reverse KL objective provides unstable gradient signals when the teacher is uncertain, preventing proper convergence.

- **Entropy-Aware On-Policy Distillation (EOPD).** We introduce an entropy-aware strategy that dynamically adapts the training objective (§4). By selectively applying reverse KL in low-entropy regions for efficiency and forward KL in high-entropy regions to preserve diversity, EOPD effectively transfers the teacher's uncertainty without the computational overhead of naive forward KL.

- **Improvements on Reasoning Benchmarks.** Empirically, EOPD maintains substantially higher generation diversity, preserving the teacher's uncertainty by retaining more probability mass in high-entropy regions than standard on-policy distillation (§5). This improvement translates into consistent downstream gains: averaged over six mathematical reasoning benchmarks, EOPD improves Avg@8 accuracy by +1.16 and Pass@8 by +1.37 for Qwen3-0.6B-Base, with larger gains of +0.99/+2.39 for the 1.7B model and +1.80/+5.05 for the 4B model.

## 2. Preliminaries

### 2.1. KL-Based Divergences

Let $P$ and $Q$ be probability distributions defined over the same sample space $\mathcal{X}$. The Kullback–Leibler (KL) divergence is a non-symmetric measure of difference between two distributions, quantifying how well $Q$ approximates the reference distribution $P$.

$$\mathrm{KL}(P \,\|\, Q) = \mathbb{E}_{x \sim P}\left[\log \frac{P(x)}{Q(x)}\right]. \quad (1)$$

Due to its asymmetry, $\mathrm{KL}(P \,\|\, Q)$ and $\mathrm{KL}(Q \,\|\, P)$ induce different optimization behaviors, depending on which distribution is used as the reference.

For auto-regressive sequence models like Large Language Models (LLMs), the probability distribution of a token $x$ is conditioned on a context $\mathbf{c}$, where $\mathbf{c}$ is the sequence generated before $x$. In distillation, we have two models, a student model denoted by $\pi_\theta(\cdot \mid \mathbf{c})$, and a teacher model $\pi_{\mathtt{te}}(\cdot \mid \mathbf{c})$. We now define forward and reverse KL divergences in terms of these models.

**Forward KL (Teacher-to-Student).** The forward KL divergence is defined as an expectation over the teacher's distribution:

$$\mathrm{KL}(\pi_{\mathtt{te}}(\cdot \mid \mathbf{c}) \,\|\, \pi_\theta(\cdot \mid \mathbf{c})) = \mathbb{E}_{x \sim \pi_{\mathtt{te}}(\cdot \mid \mathbf{c})}\left[\log \frac{\pi_{\mathtt{te}}(x \mid \mathbf{c})}{\pi_\theta(x \mid \mathbf{c})}\right]. \quad (2)$$

Minimizing (2) is equivalent to standard supervised learning (maximizing likelihood on teacher samples). It penalizes the student for assigning low probability to any token the teacher considers likely. This induces *mode-covering* behavior, where the student attempts to match the entire support of the teacher, potentially leading to overly diffuse distributions if the student has limited capacity (Minka, 2005).

**Reverse KL (Student-to-Teacher).** The reverse KL divergence is defined as an expectation over the student's distribution:

$$\mathrm{KL}(\pi_\theta(\cdot \mid \mathbf{c}) \,\|\, \pi_{\mathtt{te}}(\cdot \mid \mathbf{c})) = \mathbb{E}_{x \sim \pi_\theta(\cdot \mid \mathbf{c})}\left[\log \frac{\pi_\theta(x \mid \mathbf{c})}{\pi_{\mathtt{te}}(x \mid \mathbf{c})}\right]. \quad (3)$$

Since the expectation is taken over student samples, (3) penalizes generated tokens that the teacher considers unlikely, but ignores teacher modes that the student does not visit. It induces *mode-seeking* behavior, encouraging the student to concentrate probability mass on a single high-likelihood mode of the teacher while ignoring others (Minka, 2005).

### 2.2. On-Policy Distillation

On-policy distillation (OPD) (Agarwal et al., 2024) is a post-training method in which a student model learns by matching a teacher's token-level probability distribution on its own generated sequences. By training on on-policy rollouts rather than teacher-generated trajectories, OPD enables precise credit assignment and mitigates compounding errors inherent in off-policy imitation.

Recent OPD methods optimize the reverse KL divergence (Gu et al., 2024; Lu & Lab, 2025), encouraging mode-seeking behavior that helps the student focus on the teacher's dominant modes. Specifically, given inputs $\mathbf{q} \sim \mathcal{D}$, we denote $\mathbf{x} = (x_1, x_2, \ldots, x_{|\mathbf{x}|})$ as the student-generated token sequence. With $\mathbf{c}_t = (\mathbf{q}, x_{<t})$ as the context for token $t$, we have $x_t \sim \pi_\theta(\cdot \mid \mathbf{c}_t)$. The on-policy reverse-KL objective is then defined as:

$$\mathbb{E}_{\mathbf{q} \sim \mathcal{D}}\left[\mathbb{E}_{\mathbf{x} \sim \pi_\theta(\cdot \mid \mathbf{q})}\left[\frac{1}{|\mathbf{x}|}\sum_{t=1}^{|\mathbf{x}|} \mathcal{L}_t^{\mathrm{RKL}}(\theta; \mathbf{c}_t)\right]\right], \quad (4)$$

where $\mathcal{L}_t^{\mathrm{RKL}}(\theta; \mathbf{c}_t)$ is the per-token reverse KL from (3):

$$\mathcal{L}_t^{\mathrm{RKL}}(\theta; \mathbf{c}_t) = \mathrm{KL}(\pi_\theta(\cdot \mid \mathbf{c}_t) \,\|\, \pi_{\mathtt{te}}(\cdot \mid \mathbf{c}_t)). \quad (5)$$

In practice, Lu & Lab (2025) use a single-sample Monte-Carlo estimate of the expectation in (3) and plug it into a policy-gradient-style update. This is done by defining the token-level reward as the log-probability difference

between the teacher and student:

$$A_t = \log \pi_{\texttt{te}}(x_t \mid \mathbf{c}_t) - \log \pi_\theta(x_t \mid \mathbf{c}_t). \quad (6)$$

This reward measures how preferred the student-selected token is under the teacher distribution in the same context, assigning positive values when the teacher assigns a higher probability to the token than the student, and negative values otherwise. The student then essentially optimizes the objective $\max_\theta \mathbb{E}_{\pi_\theta}\left[\sum_t A_t\right]$. The result is an effective on-policy distillation method based on policy gradient optimization.

## 2.3. OPD with Clipped-Reverse KL

To stabilize training, the OPD objective can be implemented with standard PPO-style importance sampling and clipping (Schulman et al., 2017). We sample trajectories using a behavior policy $\pi_{\theta_{\text{old}}}$ instantiated from the student policy $\pi_\theta$, query the teacher for log-probabilities of the sampled student tokens, and define a per-token advantage $\hat{A}_t = \log \pi_{\texttt{te}}(x_t \mid \mathbf{c}_t) - \log \pi_{\theta_{\text{old}}}(x_t \mid \mathbf{c}_t)$, substituting the behavior policy in (6). We then update $\pi_\theta$ by minimizing the clipped PPO surrogate over generated tokens. To simplify notation, we omit the expectations over $\mathbf{q}$ and $\mathbf{x}$ in (4) and focus on single tokens. The surrogate objective is then

$$\mathcal{L}_t^{\text{OPD}}(\theta; \mathbf{c}_t) = \mathbb{E}_{x_t \sim \pi_{\theta_{\text{old}}}(\cdot|\mathbf{c}_t)}\left[\widetilde{A}_t\right], \quad (7)$$

where $\widetilde{A}_t$ is the clipped reverse KL loss:

$$\widetilde{A}_t = \max\left(-r_t\hat{A}_t, -\text{clip}(r_t, 1-\epsilon, 1+\epsilon)\,\hat{A}_t\right). \quad (8)$$

Here, $r_t = \frac{\pi_\theta(x_t|\mathbf{c}_t)}{\pi_{\theta_{\text{old}}}(x_t|\mathbf{c}_t)}$ corrects for sampling under $\pi_{\theta_{\text{old}}}$, while clipping limits overly large policy updates.

# 3. Diversity Degradation and Instability in On-Policy Distillation

In §3.1, we first analyze token-level entropy distributions to identify diversity degradation after on-policy distillation due to reverse KL. We then show that reverse KL produces unstable learning signals when teacher entropy is high, with the student's top-k predictions failing to converge in §3.2.

## 3.1. Token-Level Entropy Analysis

In domains that require complex reasoning, such as mathematical and multi-step reasoning tasks, high-entropy tokens[1] in the teacher model are not merely noise but encode important knowledge in the form of multiple plausible reasoning paths and meaningful uncertainty (Wang et al., 2025; Cheng et al., 2025). However, on-policy distillation may fail to properly capture this knowledge. As discussed in §2.2, it

---

[1]We use *high-entropy token* as shorthand for a token at a position where the teacher's conditional distribution has high entropy.

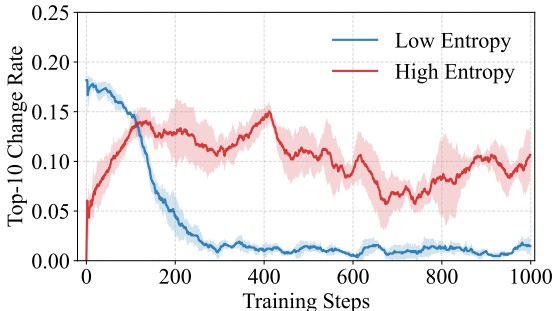

*Figure 1.* Top-10 change rate for Scenario A (blue), where the teacher distribution has low entropy, and Scenario B (red), where the teacher distribution has high entropy across 3 seeds. For Scenario B, a student optimized with reverse KL fails to capture the teacher's distribution, as evidenced by persistently high and fluctuating Top-10 change rates.

typically minimizes the reverse KL divergence, a mode-seeking objective that favors fast convergence at the cost of reduced exploration, thereby hindering the transfer of the teacher's uncertainty to the student.

To examine this effect, we generate responses from the teacher model (Qwen3-8B (Yang et al., 2025)) and an on-policy-distilled student (Qwen3-1.7B-Base) and evaluate on AIME24 and AIME25 prompts (MAA, 2025), and analyze their token entropy distributions (see §5.1 for more details on the experimental setup). We find that the distilled student retains fewer high-entropy tokens (entropy $\geq 1.0$) than the teacher, specifically only 6.8% compared to 18.5% of the teacher (see Figure 3 in §5.4 for full histogram). This suggests that reverse KL drives aggressive mode-seeking behavior rather than preserving the teacher's inherent uncertainty.

*Table 1.* Top-1 Change Count for low teacher entropy scenario and a high teacher entropy scenario. We observe that when the teacher entropy is high, the top-1 index frequently changes.

| Teacher Entropy | Temp. | Top-1 Change Count |
|:---:|:---:|:---:|
| (A) Low | 0.3 | $7.3 \pm 1.6$ |
| (B) High | 1.0 | $84.0 \pm 16.7$ |

## 3.2. Instability of Reverse KL-based Reward

To gain insight, we first conduct a simplified toy experiment to analyze how reverse KL-based reward optimization behaves under different levels of teacher uncertainty.

**Toy setup.** We study how a student learns from a teacher via reverse KL policy gradients, in a setting that retains the essential characteristics of having a teacher distribution with multiple modes and a student with limited coverage. To simplify matters, we remove the autoregressive conditioning on context $\mathbf{c}_t$. This reduces the distributions to categorical

distributions over $V$ indices (here $V = 80$) and no longer requires complex LLMs.

The teacher distribution $P_{\text{te}}$ is constructed as follows. We sample logits $\mathbf{z} \in \mathbb{R}^V$ i.i.d. from $\mathcal{N}(0,1)$, then overwrite five randomly chosen entries with larger values $\{1.7, 1.9, 2.1, 2.3, 2.5\}$ to create five modes. We then apply temperature ($T$) scaling:

$$P_{\text{te}}(x) = \text{softmax}(\mathbf{z}/T)_x.$$

We consider two scenarios: (A) $T = 0.3$, yielding a low-entropy (peaked) teacher, and (B) $T = 1.0$, yielding a high-entropy (diverse) teacher. The teacher is fixed throughout training (see Appendix B for a detailed summary and visualization of the distribution).

The student distribution $P_S$ is parameterized by learnable logits $\mathbf{s} \in \mathbb{R}^V$ initialized i.i.d. from $\mathcal{N}(0,1)$. To mimic limited model capacity, we restrict sampling to the student's top-10 indices. We denote this capacity-limited (top-10) student distribution as $P_{S^{10}}$.

**Optimization.** At each step, we sample an index $x \sim P_{S^{10}}$ and compute the reward

$$r(x) = \log P_{\text{te}}(x) - \log P_{S^{10}}(x),$$

corresponding to a sample-based estimate of the negative reverse KL. We then update the sampled logit via $s_x \leftarrow s_x + \eta r(x)$ where $\eta$ denotes the learning rate.

**Metrics.** We track two metrics during training: (1) *the top-10 change rate*, defined as the Jaccard distance (Jaccard, 1901) $\left(1 - \frac{|\mathcal{S}_t \cap \mathcal{S}_{t-1}|}{|\mathcal{S}_t \cup \mathcal{S}_{t-1}|}\right)$ between the sets of top-10 tokens $\mathcal{S}_{t-1}$, $\mathcal{S}_t$ at steps $t-1$ and $t$; and (2) *the top-1 change count*, the number of times the student's most probable index changes between updates.

**Results.** As shown in Figure 1 and Table 1, under the low-entropy teacher (Scenario A), the top-10 change rate decreases steadily and top-1 changes are rare. In contrast, under the high-entropy teacher (Scenario B), training exhibits persistent instability: the top-1 index changes frequently and the top-10 set fails to converge. These results demonstrate that reverse-KL rewards provide unstable learning signals when the teacher distribution is uncertain. This motivates training objectives that more directly transmit the teacher's distributional structure to the student.

## 4. Entropy-Aware On-Policy Distillation

To address the limitations of reverse KL in on-policy distillation, we propose Entropy-Aware On-Policy Distillation (EOPD). Our key insight is that reverse KL and forward KL offer complementary strengths: reverse KL enables efficient, stable learning on confident teacher predictions, while

**Algorithm 1** Entropy-Aware On-Policy Distillation

**Require:** Teacher policy $\pi_{\text{te}}$, student policy $\pi_\theta$
**Require:** Entropy threshold $\tau$, Top-$k$ size $k$
**Require:** PPO clip $\epsilon$, learning rate $\eta$
**Require:** Training dataset $\mathcal{D}$
1: **for** each training iteration **do**
2:     Set $\pi_{\theta_{\text{old}}} \leftarrow \pi_\theta$
3:     Sample a prompt batch $\mathcal{B} = \{\mathbf{q}_i\}_{i=1}^B \subset \mathcal{D}$
4:     Rollout buffer $\mathcal{R} \leftarrow \emptyset$
5:     **for** each prompt $\mathbf{q} \in \mathcal{B}$ **do**
6:         Sample $\mathbf{x} = (x_1, \ldots, x_{|\mathbf{x}|}) \sim \pi_{\theta_{\text{old}}}(\cdot \mid \mathbf{q})$
7:         **for** $t = 1$ to $|\mathbf{x}|$ **do**
8:             Context $\mathbf{c}_t = (\mathbf{q}, x_{<t})$
9:             Query teacher $Q_t$: $(\log \pi_{\text{te}}(x_t \mid \mathbf{c}_t), H_t^{\text{te}}, \mathcal{S}_t^k)$
10:         Store trajectory-level data:

$$\left(\mathbf{q}, \mathbf{x}, \{Q_t\}_{t=1}^{|\mathbf{x}|}\right) \rightarrow \mathcal{R}$$

11:     **for** each mini-batch gradient step **do**
12:         Sample a mini-batch of prompts $\mathcal{B}_{\text{mini}} \subset \mathcal{R}$
13:         $\mathcal{L}(\theta) = \dfrac{1}{\sum_{(\mathbf{q},\mathbf{x}) \in \mathcal{B}_{\text{mini}}} |\mathbf{x}|} \displaystyle\sum_{(\mathbf{q},\mathbf{x}) \in \mathcal{B}_{\text{mini}}} \sum_{t=1}^{|\mathbf{x}|} \mathcal{L}_t^{\text{EOPD}}(\theta; \mathbf{c}_t)$
        using (9)
14:         Update parameters: $\theta \leftarrow \theta - \eta \nabla_\theta \mathcal{L}(\theta)$

forward KL's mode-covering property transfers the teacher's uncertainty and global structure. However, naively applying forward KL forces the student to cover the teacher's full distribution, including low-probability tails. This can degrade training efficiency, especially for students with limited capacity (Gu et al., 2024; Cha & Cho, 2025).

To leverage the best of both objectives, we propose an *entropy-aware* strategy that selectively applies forward KL based on the teacher's token-level uncertainty. Specifically, we define our token-level objective as:

$$\mathcal{L}_t^{\text{EOPD}}(\theta; \mathbf{c}_t) = \mathcal{L}_t^{\text{OPD}}(\theta; \mathbf{c}_t) + \alpha \cdot \mathbb{I}\big[H_t^{\text{te}} > \tau\big] \mathcal{L}_t^{\text{FKL}}(\theta; \mathbf{c}_t), \quad (9)$$

where $H_t^{\text{te}} = -\sum_{x \in \mathcal{V}} \pi_{\text{te}}(x|\mathbf{c}_t) \log \pi_{\text{te}}(x|\mathbf{c}_t)$ denotes the teacher's token-level entropy at position $t$, $\mathcal{V}$ denotes the vocabulary, and the forward KL divergence is

$$\mathcal{L}_t^{\text{FKL}}(\theta; \mathbf{c}_t) = \text{KL}(\pi_{\text{te}}(\cdot \mid \mathbf{c}_t) \,\|\, \pi_\theta(\cdot \mid \mathbf{c}_t)).$$

The first term $\mathcal{L}_t^{\text{OPD}}(\theta; \mathbf{c}_t)$ in (9) corresponds to the clipped reverse KL loss defined in (8). The second term $\mathcal{L}_t^{\text{FKL}}(\theta; \mathbf{c}_t)$ is activated only when $H_t^{\text{te}} > \tau$, encouraging the student to preserve probability mass over multiple plausible continuations. In low-entropy regions where the teacher is confident, the objective reduces to standard reverse KL, retaining its efficiency and fast convergence. In high-entropy regions, forward KL prevents mode collapse and preserves the teacher's

distributional diversity. The hyperparameters $\tau$ and $\alpha$ control this transition; we study their effects in Appendix I. In our experiments, we use $\tau = 0.8$ and $\alpha = 1.0$.

The full objective function for EOPD is like (4), where we bring back the expectations over prompts $\mathbf{q}$ and generated tokens $\mathbf{x}$, but with $\mathcal{L}_t^{\text{RKL}}(\theta; \mathbf{c}_t)$ replaced by $\mathcal{L}_t^{\text{EOPD}}(\theta; \mathbf{c}_t)$ in (9). We use Algorithm 1 to optimize this objective. The expectations over $\mathbf{q}$ and $\mathbf{x}$ are approximated by sampling batches $\mathcal{B}$ (line 3 in Algorithm 1) and generating rollouts using the behavior policy $\pi_{\text{old}}$ (line 6). The teacher is then queried to collect quantities needed to compute the objective (line 9). For the $\mathcal{L}_t^{\text{OPD}}(\theta; \mathbf{c}_t)$ term in (9), the expectation in (7) is approximated by a single-sample Monte Carlo estimate as discussed previously. In effect, the clipped reverse KL loss (8) is evaluated at the token $x_t$ sampled during the rollouts. The forward KL $\mathcal{L}_t^{\text{FKL}}(\theta; \mathbf{c}_t)$ is approximated not by sampling, but as an expectation computed over the teacher's top-$k$ tokens $\mathcal{S}_t^k$:

$$\mathcal{L}_t^{\text{FKL}}(\theta; \mathbf{c}_t) \approx \sum_{x \in \mathcal{S}_t^k} \tilde{\pi}_{\text{te}}(x \mid \mathbf{c}_t) \log \frac{\tilde{\pi}_{\text{te}}(x \mid \mathbf{c}_t)}{\pi_\theta(x \mid \mathbf{c}_t)}, \quad (10)$$

where $\tilde{\pi}_{\text{te}}(\cdot \mid c_t)$ denotes the teacher distribution renormalized over the top-$k$ tokens $\mathcal{S}_t^k$, defined as

$$\tilde{\pi}_{\text{te}}(x \mid \mathbf{c}_t) = \frac{\pi_{\text{te}}(x \mid \mathbf{c}_t)}{\sum_{x' \in \mathcal{S}_t^k} \pi_{\text{te}}(x' \mid \mathbf{c}_t)}, \qquad x \in \mathcal{S}_t^k.$$

We restrict to the teacher's top-$k$ tokens to ensure that the student does not have to learn from the low-probability tails of the teacher, along with improving computational efficiency (Shum et al., 2024; Peng et al., 2025). In Appendix F, we analyze the tradeoff between cumulative probability mass and memory cost across different $k$, and show that $k = 16$ offers a practical balance between the two.

By adapting to the teacher's local uncertainty, EOPD balances training stability with diversity preservation, enabling effective knowledge transfer across both confident and ambiguous regions of the output distribution.

## 5. Experiments

We address the following research questions:

**RQ1:** Does EOPD improve mathematical reasoning performance compared to existing baselines? (§5.2)

**RQ2:** Does EOPD improve out-of-domain performance compared to existing baselines? (§5.3)

**RQ3:** How does EOPD affect the transfer of token-level uncertainty from the teacher to the student? (§5.4)

**RQ4:** How does EOPD differ from other entropy-promoting methods in transferring teacher uncertainty? (§5.5)

### 5.1. Experimental Settings

**Models and Training Datasets.** We conduct experiments using three Qwen3 (Yang et al., 2025) student models of different sizes, Qwen3-0.6B-Base, Qwen3-1.7B-Base, and Qwen3-4B-Base. We use Qwen3-8B as the teacher model, without enabling thinking mode. For the 0.6B and 1.7B student models, training was performed using the MATH (Hendrycks et al., 2021) dataset, while for the 4B student model, the more challenging DAPO (Yu et al., 2025) dataset is used.

**Evaluation Benchmarks and Metrics.** We evaluate our models on MATH500 (Hendrycks et al., 2021), AIME24/25 (MAA, 2025), AMC23 (MAA, 2023), Minerva (Lewkowycz et al., 2022), and OlympiadBench (He et al., 2024), using a rollout temperature of 1.0, top-$p$ sampling with $p = 0.8$, and a maximum response length of 8192 tokens. We sample 8 responses per question and report the average accuracy (Avg@8) and pass rate (Pass@8).

**Baselines.** We compare our method with several baselines:

1. **Knowledge Distillation** (Hinton et al., 2015; Kim & Rush, 2016): KD trains the student model by minimizing forward KL divergence with respect to the teacher's distribution, along with a cross-entropy loss on hard labels from an off-policy dataset generated by the teacher.

2. **On-Policy Distillation** (Lu & Lab, 2025): The student is trained using on-policy trajectories, following the formulation described in §2.3.

3. **Group Relative Policy Optimization (GRPO)** (Shao et al., 2024): GRPO optimizes the policy by comparing verifiable rewards across multiple sampled outputs for the same input.

**Implementation.** For off-policy distillation (KD), we utilize the DistillKit (Goddard & Atkins, 2024) framework. On-policy distillation and GRPO are implemented using the verl (Sheng et al., 2024) framework. We train with a batch size of $B = 128$ and a mini-batch size of $B_{\text{mini}} = 32$, resulting in 4 gradient steps per training iteration. Full implementation details are provided in Appendix A.

### 5.2. Main Results

**Performance on Mathematical Reasoning.** As shown in Table 2, EOPD demonstrates stable performance improvements across six mathematical reasoning benchmarks, achieving improved or competitive results in terms of Avg@8 and Pass@8. In particular, compared to OPD, EOPD improves Avg@8 by +1.16 and Pass@8 by +1.37 on average across the six benchmarks for the Qwen3-0.6B-Base model, Avg@8 by +0.99 and Pass@8 by +2.39 for the Qwen3-1.7B-Base model, and +1.80 in Avg@8 and +5.05 in Pass@8 for the Qwen3-4B-Base model. These improve-

*Table 2.* Main results (accuracy %) on six mathematical reasoning benchmarks. EOPD demonstrates consistent improvements across benchmarks and model scales. Parentheses indicate training data. **Bold** indicates best performance and underline indicates second-best.

| Benchmark | Metric | Qwen3-0.6B-Base (MATH) | | | | Qwen3-1.7B-Base (MATH) | | | | Qwen3-4B-Base (DAPO-Math-14k) | | | |
|---|---|---|---|---|---|---|---|---|---|---|---|---|---|
| | | KD | GRPO | OPD | EOPD (ours) | KD | GRPO | OPD | EOPD (ours) | KD | GRPO | OPD | EOPD (ours) |
| MATH500 | Avg@8 | 47.80 | 51.83 | 50.09 | **52.02** | 63.80 | **68.83** | 67.76 | 68.73 | 74.73 | **80.47** | 78.81 | 80.20 |
| | Pass@8 | 69.60 | 74.40 | 73.20 | **76.00** | 84.20 | 84.60 | 84.80 | **87.60** | 92.20 | 91.00 | 90.80 | **93.00** |
| AMC23 | Avg@8 | 23.43 | 25.25 | 24.69 | **27.81** | 38.11 | 40.62 | 39.06 | **41.88** | 48.13 | 58.44 | 57.33 | **60.94** |
| | Pass@8 | 52.50 | 55.00 | **57.50** | 55.00 | 70.00 | 70.00 | 70.00 | **75.00** | **85.00** | 75.00 | 80.00 | **87.50** |
| Minerva | Avg@8 | 18.61 | 20.08 | 19.98 | **20.82** | 28.14 | 29.46 | 29.83 | **30.15** | 34.65 | **40.81** | 40.08 | 39.71 |
| | Pass@8 | 36.03 | 35.66 | 34.93 | **37.13** | 44.10 | 48.16 | 47.06 | **50.74** | 55.15 | 55.51 | 54.00 | **56.99** |
| OlympiadBench | Avg@8 | 18.15 | 18.91 | **20.15** | 19.52 | 27.66 | 29.89 | 30.09 | **30.28** | 37.33 | **43.37** | 42.10 | 43.24 |
| | Pass@8 | 36.59 | 34.96 | 37.19 | **39.56** | 48.40 | 50.81 | **51.56** | 51.11 | 62.52 | 60.74 | 58.80 | **63.11** |
| AIME24 | Avg@8 | 2.19 | **4.58** | 2.50 | 4.17 | 10.06 | 9.17 | 8.33 | **10.42** | 12.50 | 14.85 | **18.33** | 17.92 |
| | Pass@8 | 6.67 | 10.00 | 10.00 | **13.33** | 20.00 | 20.00 | 20.00 | **23.33** | 30.00 | 30.00 | 26.67 | **36.67** |
| AIME25 | Avg@8 | 0.83 | 0.83 | 1.25 | **1.25** | 3.36 | 4.58 | **6.25** | 5.83 | 12.08 | 12.66 | 12.08 | **17.50** |
| | Pass@8 | 6.67 | **10.00** | 6.67 | 6.67 | 16.67 | 16.67 | 16.67 | 16.67 | 23.33 | 26.67 | 30.00 | **33.33** |

ments highlight the effectiveness of EOPD, particularly its ability to better transfer the teacher's local uncertainty.

**Pass@$k$ Performance.** Pass@$k$ measures the probability of obtaining at least one correct solution among $k$ sampled rollouts. While Avg@$k$ reflects average reasoning quality, Pass@$k$ more directly captures the model's problem-solving capability by approximating best-case performance under multiple samples. In Figure 2, we report Pass@$k$ for AIME24 and AIME25 with $k$ ranging from 8 to 128, and for AMC23 with $k$ ranging from 4 to 64. EOPD achieves consistently higher Pass@k compared to OPD. Notably, on harder benchmarks such as AIME, the gap between the two methods widens as $k$ increases. This suggests that EOPD more effectively explores diverse reasoning trajectories, thereby increasing the likelihood of reaching a correct solution. See Appendix D for more results on Pass@k performance.

*Table 3.* Out-of-domain benchmark results on Qwen3-1.7B-Base, covering general reasoning and instruction following. Our method achieves higher average accuracy and pass@8 on general reasoning, and is competitive or superior in win rate (*WR*) and length-controlled win rate (*LC-WR*). Best and second-best results are shown in **bold** and underline, respectively.

| Benchmark | Metric | KD | GRPO | OPD | EOPD |
|---|---|---|---|---|---|
| GPQA-Diamond | Avg@8 | 27.01 | 27.86 | 30.08 | **31.50** |
| | Pass@8 | 75.20 | 77.83 | 77.78 | **81.31** |
| MMLU-Pro | Pass@1 | 37.54 | 41.46 | 42.26 | **43.20** |
| AlpacaEval 2.0 | LC-WR | 19.30 | 16.86 | 22.86 | **23.83** |
| | WR | 23.63 | 20.55 | **29.92** | 29.54 |

### 5.3. Out-of-domain Evaluation

Table 3 reports performance on out-of-domain benchmarks for the Qwen3-1.7B-Base student: GPQA-Diamond (Rein et al., 2024), MMLU-Pro (Wang et al., 2024), and AlpacaEval 2.0 (Dubois et al., 2024), which evaluate general reasoning and instruction-following abilities. Although trained exclusively on math data, OPD and EOPD exhibit stable

performance increases over KD and GRPO across these benchmarks, indicating that on-policy distillation transfers useful reasoning behaviors beyond the training distribution.

Furthermore, EOPD outperforms OPD on all out-of-domain benchmarks except for AlpacaEval 2.0 win rate. This suggests that selectively incorporating teacher guidance on high-uncertainty tokens also provides additional benefits for general reasoning and instruction-following.

### 5.4. Token-Level Entropy Analysis

To support the hypothesis that EOPD more effectively explores diverse reasoning trajectories, we conduct a token-level analysis of uncertainty transfer from the Qwen3-8B teacher to the Qwen3-1.7B-Base student. Following §3.1, we compare students trained with OPD and EOPD on the AIME24 and AIME25 benchmarks.

For each generated token, we compute the entropy of the model's predicted distribution conditioned on the preceding context. Figure 3 aggregates these values into a histogram. In the mid-entropy range (approximately 0.1–1.0), OPD and EOPD exhibit similar distributions and remain close to the teacher. The key difference emerges at higher entropy values (entropy $\geq 1.0$): EOPD retains substantially more probability mass in this region and stays much closer to the teacher, whereas OPD markedly under-represents it.

High-entropy tokens correspond to intrinsically ambiguous decision points where the teacher assigns non-negligible probability to multiple plausible continuations. Therefore, preserving a teacher-like distribution at these positions is likely important. We hypothesize that EOPD's improved preservation of high-entropy tokens mitigates premature over-confidence and mode collapse, contributing to the downstream performance gains observed in §5.2.

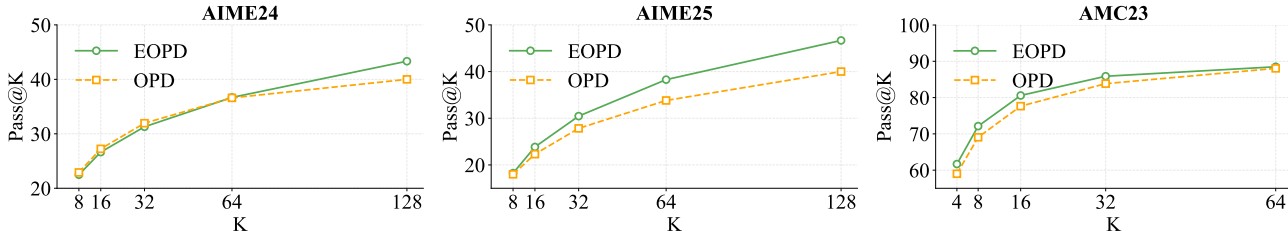

*Figure 2.* Pass@$k$ performance comparison between OPD and EOPD on the AIME and AMC benchmarks with the Qwen3-1.7B-Base student. EOPD achieves higher Pass@$k$ compared to OPD, with the performance gap becoming more pronounced as $k$ increases.

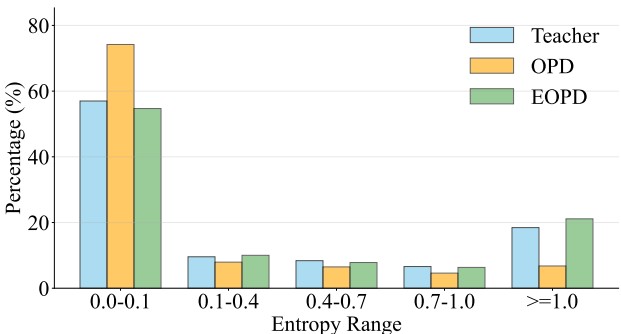

*Figure 3.* Token-level entropy histograms comparing the Qwen3-8B teacher with Qwen3-1.7B-Base trained using OPD and EOPD on the AIME 2024 and 2025 benchmarks. While both methods exhibit similar distributions to the teacher in the mid-entropy range, EOPD preserves more probability mass in the high-entropy region, staying closer to the teacher than OPD.

*Figure 4.* Average policy entropy during training for the Qwen3-1.7B-Base student trained with OPD + Entropy Bonus, OPD + Advantage Shaping, and EOPD. Advantage Shaping converges to a lower entropy regime, while Entropy Bonus maintains entropy levels comparable to EOPD.

### 5.5. Comparison with Entropy-Driven Baselines

We compare our method with entropy-based approaches commonly used to encourage exploration in reinforcement learning. Specifically, we evaluate the Qwen3-1.7B-Base student using two strategies: an entropy bonus (Schulman et al., 2017), which explicitly regularizes the policy toward higher entropy, and advantage shaping (Cheng et al., 2025), which augments the advantage with an entropy-dependent term to bias updates toward high-uncertainty actions. Detailed implementation details are provided in Appendix A.

As shown in Table 4, EOPD outperforms both baselines across several benchmarks. To explain these gains, we analyze entropy dynamics (Figure 4) and forward KL at high-entropy positions (Figure 5). OPD + Advantage Shaping exhibits substantially lower training entropy than other methods, indicating limited diversity and explaining its poor performance. However, EOPD and OPD + Entropy Bonus maintain similar entropy levels, so entropy alone does not explain EOPD's advantage. Looking at the forward KL at positions where the teacher's entropy exceeds $\tau = 0.8$, EOPD achieves consistently lower values than OPD + Entropy Bonus, indicating better alignment with the teacher in uncertain regions. Overall, these results show that preserving entropy alone is insufficient for the student to match the

teacher's diversity, especially in high-entropy regions.

### 5.6. Comparison with GKD

We compare EOPD with GKD (Agarwal et al., 2024), which optimizes a divergence objective over a mixture of off-policy data and on-policy rollouts controlled by a coefficient $\lambda$. Following GKD, we use forward KL for reasoning experiments. Specifically, the off-policy data is generated by the teacher model (Qwen3-8B), while the on-policy data is sampled from the student (Qwen3-1.7B-Base) during training. We evaluate $\lambda \in \{0.5, 0.8, 1.0\}$. We observe that the fully on-policy setting ($\lambda = 1.0$) achieves the best performance among different $\lambda$ values, consistent with GKD's observation that on-policy rollouts are particularly important for reasoning tasks. As shown in Table 5, EOPD consistently outperforms the tuned GKD baseline across all benchmarks.

### 5.7. Effect of KL Objective

To analyze the effect of different KL objectives, we compare EOPD with OPD and OPD-FKL using the Qwen3-1.7B-Base student. Here, OPD optimizes the policy using the reverse KL-based reward discussed throughout the paper, while OPD-FKL optimizes the policy using only forward

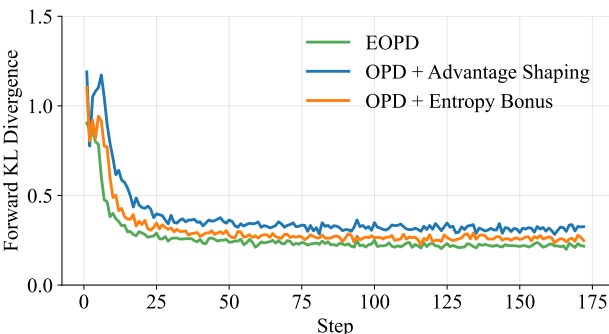

*Figure 5.* Average forward KL divergence measured during training at token positions where the teacher distribution exhibits high entropy (entropy $\geq 0.8$) for the Qwen3-1.7B-Base student. Compared to OPD + Entropy Bonus and OPD + Advantage Shaping, EOPD maintains lower forward KL values throughout training, indicating closer alignment with the teacher distribution in regions of high uncertainty.

*Table 4.* Comparison with entropy-driven exploration baselines for the Qwen3-1.7B-Base student. EOPD achieves higher Avg@8 and Pass@8 compared to OPD + Entropy Bonus and OPD + Advantage Shaping. **Bold** indicates best performance and underline indicates second-best.

| Benchmark | Metric | Entropy Bonus | Advantage Shaping | EOPD |
|---|---|---|---|---|
| MATH500 | Avg@8 | 66.87 | 67.90 | **68.73** |
|  | Pass@8 | 86.00 | 85.20 | **87.60** |
| AMC23 | Avg@8 | 39.69 | 36.56 | **41.88** |
|  | Pass@8 | **75.00** | **75.00** | **75.00** |
| AIME24 | Avg@8 | 9.58 | 8.75 | **10.42** |
|  | Pass@8 | 20.00 | **23.33** | **23.33** |
| AIME25 | Avg@8 | 4.58 | **5.83** | **5.83** |
|  | Pass@8 | 16.67 | **20.00** | 16.67 |

KL under the same on-policy setup. As shown in Table 6, reverse KL-based OPD generally outperforms OPD-FKL across most benchmarks, while EOPD outperforms both. These results suggest that reverse KL is effective for capturing the dominant modes of the teacher distribution, while selectively applying forward KL in regions with high teacher uncertainty is important for further improving performance.

### 5.8. Impact of Forward KL Placement

To study the impact of token selection for application of forward KL, we compare EOPD against two alternative forward KL augmentation strategies for the Qwen3-1.7B-Base student: full forward KL, applied at all token positions, and random forward KL, applied to a randomly selected 20% of positions[2]. As shown in Table 7, EOPD shows competitive performance compared to full and random forward

---

[2]This choice is motivated by experiments with EOPD at $\tau = 0.8$. As shown in Figure 9, forward KL is applied to approximately 15–20% of tokens on average.

*Table 5.* Comparison with GKD baselines for the Qwen3-1.7B-Base student. We evaluate GKD with different $\lambda$ values following the original setup. EOPD consistently outperforms all GKD variants across reasoning benchmarks. **Bold** indicates best performance and underline indicates second-best.

| Benchmark | Metric | GKD ($\lambda = 0.5$) | GKD ($\lambda = 0.8$) | GKD ($\lambda = 1.0$) | EOPD |
|---|---|---|---|---|---|
| MATH500 | Avg@8 | 64.53 | 65.28 | 67.20 | **68.73** |
|  | Pass@8 | 84.40 | 84.20 | 85.00 | **87.60** |
| AMC23 | Avg@8 | 35.62 | 37.50 | 38.44 | **41.88** |
|  | Pass@8 | 67.50 | 70.00 | 67.50 | **75.00** |
| AIME24 | Avg@8 | 5.83 | 6.67 | 9.17 | **10.42** |
|  | Pass@8 | 10.00 | 13.33 | **23.33** | **23.33** |
| AIME25 | Avg@8 | 4.58 | 3.75 | 5.00 | **5.83** |
|  | Pass@8 | 10.00 | 10.00 | 13.33 | **16.67** |

*Table 6.* Comparison of different KL objectives for the Qwen3-1.7B-Base student. EOPD outperforms both OPD and OPD-FKL across most benchmarks, while OPD generally outperforms OPD-FKL. **Bold** indicates the best performance and underline indicates second-best.

| Benchmark | Metric | OPD-FKL | OPD | EOPD |
|---|---|---|---|---|
| MATH500 | Avg@8 | 67.20 | 67.76 | **68.73** |
|  | Pass@8 | 85.00 | 84.80 | **87.60** |
| AMC23 | Avg@8 | 38.44 | 39.06 | **41.88** |
|  | Pass@8 | 67.50 | 70.00 | **75.00** |
| AIME24 | Avg@8 | 9.17 | 8.33 | **10.42** |
|  | Pass@8 | **23.33** | 20.00 | **23.33** |
| AIME25 | Avg@8 | 5.00 | **6.25** | 5.83 |
|  | Pass@8 | 13.33 | **16.67** | **16.67** |

KL. Notably, random forward KL underperforms the other two approaches across most benchmarks, suggesting that the placement of forward KL on appropriate positions, like EOPD is important.

## 6. Related Work

**Knowledge Distillation** (Buciluǎ et al., 2006; Hinton et al., 2015) trains a smaller model to approximate a larger model's output distribution. For auto-regressive models, various approaches have been proposed, including matching token-level distributions (Sanh et al., 2019), teacher-generated sequences (Kim & Rush, 2016), attention scores (Wang et al., 2020), and alternative divergence objectives (Wen et al., 2023; Ko et al., 2024; Shing et al., 2025) to overcome limitations of forward and reverse KL. To address exposure bias from the mismatch between teacher-generated training data and self-generated sequences at inference (Bengio et al., 2015; Ranzato et al., 2015), several on-policy methods have been proposed, including combining off-policy and on-policy data (Lin et al., 2020; Agarwal et al., 2024), reverse KL over student-generated contexts (Gu et al., 2024;

*Table 7.* Comparison with different forward KL placement strategies for the Qwen3-1.7B-Base student. EOPD shows competitive performance against full and random forward KL, while random forward KL underperforms the other two approaches. **Bold** indicates the best performance and underline indicates second-best.

| Benchmark | Metric | Full | Random | EOPD |
|---|---|---|---|---|
| MATH500 | Avg@8 | 67.58 | 67.33 | **68.73** |
|  | Pass@8 | 86.20 | 84.80 | **87.60** |
| AMC23 | Avg@8 | 39.39 | 36.88 | **41.88** |
|  | Pass@8 | **75.00** | 67.50 | **75.00** |
| AIME24 | Avg@8 | 8.75 | 7.92 | **10.42** |
|  | Pass@8 | **23.33** | 20.00 | **23.33** |
| AIME25 | Avg@8 | 5.00 | 5.42 | **5.83** |
|  | Pass@8 | **20.00** | 13.33 | 16.67 |

Lu & Lab, 2025), and interleaved sampling (Xu et al., 2024). In addition, approaches that adapt the divergence objective based on teacher-student discrepancy measures, such as entropy gaps (Amara et al., 2022), head-tail mismatch (Wu et al., 2025), and vocabulary-level discrepancy (Jung et al., 2025), have also been explored. Our work specifically focuses on the instability and diversity degradation caused by reverse KL in high-entropy regions under on-policy distillation, selectively applying forward KL based on teacher uncertainty.

**Reasoning Abilities of Language Models** have advanced through prompting (Wei et al., 2022; Yao et al., 2023), test-time scaling (Muennighoff et al., 2025; Snell et al., 2024), and distillation from larger models (Guha et al., 2025; He et al., 2025; Guo et al., 2025). Recently, RL methods have received particular attention, including methods that optimize verifiable rewards (Shao et al., 2024; Yu et al., 2025; Liu et al., 2025) or apply step-level supervision using process reward models (Uesato et al., 2022; Lightman et al., 2023). In addition, Cheng et al. (2025); Wang et al. (2025) have proposed entropy-based methods to encourage exploration during training. Instead of relying on intrinsic entropy for exploration, our method uses teacher-guided KL selection to transfer uncertainty and distributional structure.

## 7. Conclusion

We study on-policy distillation for language models and identify a key limitation of the reverse-KL objective: its mode-seeking nature can collapse generation diversity and destabilize learning at token positions where the teacher distribution has high entropy. To address this, we introduce EOPD, which adapts the on-policy distillation objective and selectively applies forward-KL for high-entropy teacher tokens. Empirically, EOPD better preserves the teacher-like distribution while retaining on-policy training efficiency, yielding consistent gains over standard on-policy distilla-

tion on six mathematical reasoning benchmarks. Overall, our results demonstrate that explicitly modeling teacher uncertainty is essential for stable, diverse, and effective knowledge transfer.

## Acknowledgements

This work was supported by Institute for Information & communications Technology Planning & Evaluation(IITP) grant funded by the Korea government(MSIT) (RS-2019-II190075, Artificial Intelligence Graduate School Program(KAIST)), the MSIT(Ministry of Science, ICT), Korea, under the Global Research Support Program in the Digital Field program(RS-2024-00436680) supervised by the IITP(Institute for Information & Communications Technology Planning & Evaluation), and the Institute of Information & Communications Technology Planning & Evaluation(IITP) grant funded by the Korea government(MSIT) (RS-2025-02304967, AI Star Fellowship(KAIST))

## Impact Statement

This work improves on-policy knowledge distillation for transferring reasoning capabilities from large language models to smaller, more efficient models. We identify that reverse KL–based distillation, while effective for learning dominant modes, fails to preserve distributional structure and diversity in high-uncertainty regions that are critical for reasoning. To address this, we propose an entropy-aware framework that adaptively applies KL objectives based on the teacher's token-level uncertainty. The proposed method preserves the efficiency of on-policy distillation while more faithfully transferring the teacher's uncertainty and global structure. More broadly, this work contributes to the development of efficient and deployable language models, which may reduce computational and environmental costs associated with large-scale model deployment. As with prior advances in language modeling, the societal impact of this work depends on downstream applications, with no new ethical concerns beyond those of existing language models.

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

## A. Implementation Details

**Off-policy Training.** We perform off-policy distillation using DistillKit (Goddard & Atkins, 2024). The hyperparameters for all off-policy distillation experiments are summarized in Table 8. In this setting, we first sample a response from the teacher model for each question, and then train the student model on the resulting teacher context using a combination of cross-entropy loss and forward KL loss.

*Table 8.* Hyperparameters used for Off-policy distillation.

| Hyperparameter | Value |
|---|---|
| Learning rate | 1e-5 |
| LR scheduler type | cosine |
| Optimizer | AdamW |
| CE loss weight | 0.5 |
| KL loss weight | 0.5 |
| Training Batch size | 128 |
| Training epoch | 3 |
| Cutoff length | 4096 |
| Top-$k$ (for FKL) | 16 |

**On-policy Training.** On-policy distillation and GRPO are implemented using the `verl` (Sheng et al., 2024) framework. As shown in Table 9, we use a batch size of $B = 128$ and a mini-batch size of $B_{\text{mini}} = 32$, which results in four gradient update steps per training iteration. For on-policy distillation, we generate a single rollout per problem during training. In contrast, for GRPO, we generate eight rollouts per problem to enable relative comparison among trajectories.

*Table 9.* Hyperparameters used for On-policy distillation and GRPO.

| Hyperparameter | OPD, EOPD | GRPO |
|---|---|---|
| Learning rate | 3e-6 | 3e-6 |
| LR scheduler type | cosine | cosine |
| Optimizer | AdamW | AdamW |
| Training batch size | 128 | 128 |
| Mini batch size | 32 | 32 |
| Samples per prompt | 1 | 8 |
| Top-$p$ | 1.0 (Qwen), 0.8 (Llama) | 1.0 (Qwen), 0.8 (Llama) |
| Max response length | 4096 | 4096 |
| Top-$k$ (for FKL) | 16 | - |
| Training temperature | 1.0 | 1.0 |
| Training epoch | 3 (MATH), 2 (DAPO) | 3 (MATH), 2 (DAPO) |

**Computational Overhead and Efficiency Analysis.** All experiments were conducted on a $4\times$A100 80GB GPU setup. In addition, to analyze the additional computational overhead introduced by EOPD, we measure the cost of each component separately as follows.

1. **Teacher Query.** EOPD does not require additional teacher forward passes compared to OPD. Both methods query the teacher once for each student-generated token position.

2. **Training Time.** The additional cost of EOPD mainly comes from top-k extraction, renormalization, and forward KL loss computation. Empirically, EOPD introduces an average overhead of 2.16 seconds per training step, which corresponds to approximately 4.5% of the average step time (47.8 seconds). Since most of the computation time is spent on student on-policy generation ($\sim$37.7 seconds), the additional overhead remains limited.

3. **Memory Usage.** For forward KL computation, tensors corresponding to selected tokens are stored per microbatch, resulting in an average additional memory usage of 219.3 MB under selective application. In contrast, applying forward KL to all token positions requires an average of 1617.5 MB of additional memory. This demonstrates that selective entropy-aware routing is approximately $7.37\times$ more memory-efficient than full forward KL application.

**Chat Template.** During model training, for knowledge distillation, we used the same chat template as the teacher model to effectively learn the teacher distribution. Since the teacher model used in our experiments was the Qwen3-Instruct (Yang et al., 2025) model in non-thinking mode, the chat template was defined as follows:

```
<|im_start|>user\n{query}<|im_end|>\n<|im_start|>assistant\n<think>\n\n</think>
```

For GRPO training, we used the default chat template of the Qwen3-Base model. The template used is as follows:

```
<|im_start|>user\n{query}<|im_end|>\n<|im_start|>assistant\n
```

**Entropy Bonus.** Entropy Bonus (Schulman et al., 2017) adds the policy entropy directly to the optimization objective, discouraging premature policy convergence and encouraging stochastic action selection during training.

**Advantage Shaping.** Advantage Shaping (Cheng et al., 2025) augments the advantage function by adding a gradient-detached entropy term shaped by $\psi(\cdot)$:

$$A_t \leftarrow A_t + \psi(\mathcal{H}_t).$$

This mechanism encourages the model to reinforce uncertain decisions when $A_t > 0$ by providing additional reward, while reducing the penalty for uncertain decisions when $A_t < 0$, thereby promoting more exploratory reasoning behavior.

Here, the shaping function $\psi(\cdot)$ is defined as

$$\psi\left(\mathcal{H}_t\right) = \min\left(\alpha\mathcal{H}_t^{\text{detach}}, \frac{|A_t|}{\kappa}\right), \tag{11}$$

where $\alpha > 0$ and $\kappa > 1$ are hyperparameters. In the experiment described in §5.5, we follow the setup of (Cheng et al., 2025) and set $\alpha = 0.1$ and $\kappa = 2$.

# B. Toy Experiment Visualizations

In this section, we present visualizations of the toy experiment under each scenario. As shown in Figure 6, when the teacher distribution exhibits low entropy, the student model converges closely to the teacher distribution. However, when the teacher distribution has high entropy, optimization using a reverse-KL-based reward causes the student to concentrate on a small number of dominant indices. As a result, the global structure of the teacher distribution is not fully captured.

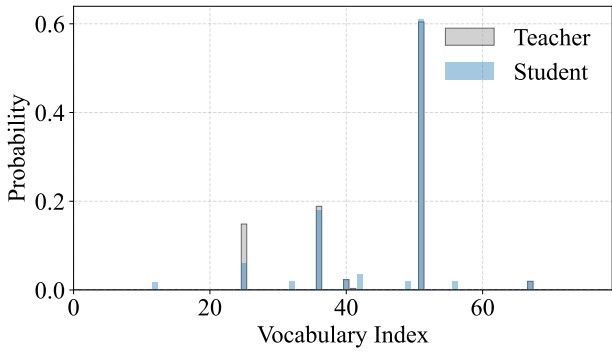 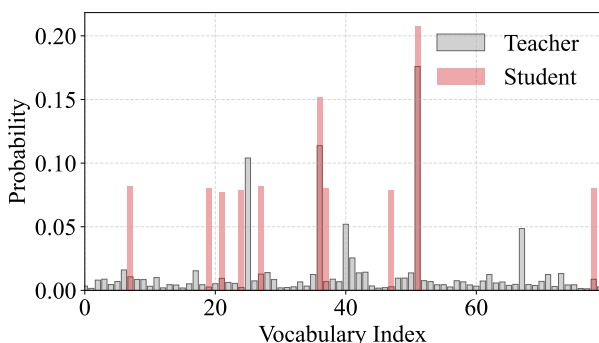

*(a)* Low Entropy Teacher Distribution         *(b)* High Entropy Teacher Distribution

*Figure 6.* Teacher–student distributions under low and high-entropy scenarios. When the teacher distribution has low entropy, the student model accurately converges to the teacher. In contrast, when the teacher distribution has high entropy, optimization with reverse KL based reward leads the student to align with a few dominant indices of the teacher distribution, while the global structure is not fully approximated.

## C. Evaluation Details

Math reasoning benchmarks (six datasets) and the GPQA-Diamond (Rein et al., 2024) evaluation were conducted using the evaluation pipeline released with Qwen2.5-Math (Yang et al., 2024). All experiments were performed in a zero-shot setup. During sampling, the maximum sequence length was set to 8192, with temperature = 1.0 and top-p = 0.8. We report both average@k and pass@k as evaluation metrics. For all problems, the same prompt template was appended at the end: ``Please reason step by step, and put your final answer within \boxed{}.''

For out-of-domain (OOD) evaluation, MMLU-Pro (Wang et al., 2024) was evaluated under the default 5-shot setting (examples are sampled from the validation set), while all other configurations were kept identical to those used in the math reasoning benchmarks. The prompt template for MMLU-Pro is specified below. For AlpacaEval (Dubois et al., 2024), we followed the default evaluation protocol and used the annotator model `weighted_alpaca_eval_gpt4_turbo` to report both win rate and length-controlled win rate against gpt4_turbo (Achiam et al., 2023).

```
The following are multiple choice questions (with answers) about {$}.  Think
step by step and then finish your answer with ``the answer is (X)'' where X
is the correct letter choice.
```

## D. Pass@$k$ Experiments

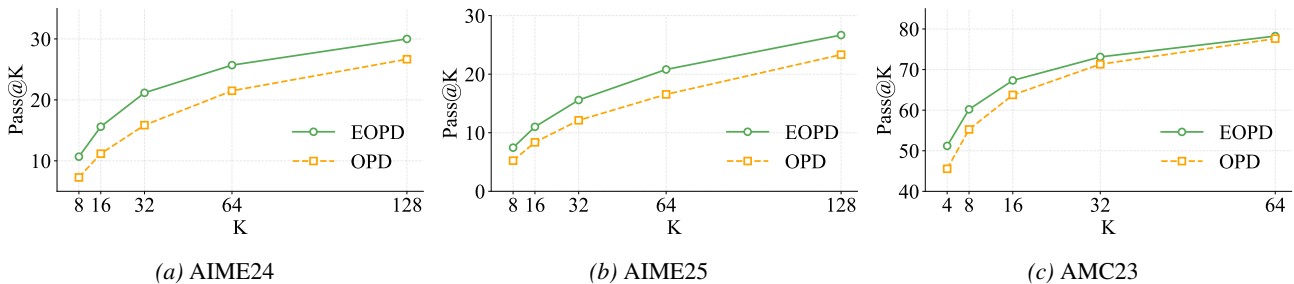

*(a)* AIME24                     *(b)* AIME25                     *(c)* AMC23

*Figure 7.* Pass@$k$ performance comparison between OPD and EOPD on the AIME and AMC benchmarks with the Qwen3-0.6B-Base student. EOPD achieves higher Pass@$k$ compared to OPD.

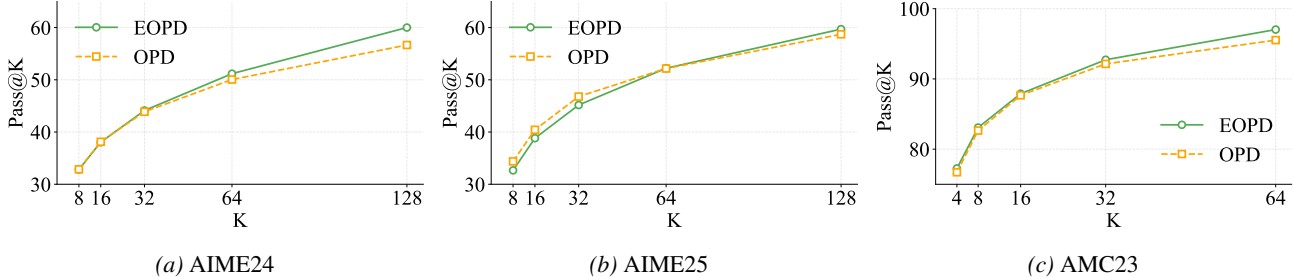

*(a)* AIME24                     *(b)* AIME25                     *(c)* AMC23

*Figure 8.* Pass@$k$ performance comparison between OPD and EOPD on the AIME and AMC benchmarks with the Qwen3-4B-Base student. EOPD achieves higher Pass@$k$ compared to OPD, with the performance gap becoming more pronounced as $k$ increases.

Figure 7 and Figure 8 show the Pass@$k$ performance across different values of $k$ for the Qwen3-0.6B-Base and Qwen3-4B-Base models. Compared to OPD, EOPD achieves better performance on AIME24/25 (MAA, 2025) and AMC23 (MAA, 2023). On harder benchmarks such as AIME, the performance gap between the two methods widens as $k$ increases. This suggests that EOPD more effectively explores diverse reasoning trajectories, thereby increasing the likelihood of reaching a correct solution.

# E. High Entropy Token Ratio

Figure 9 shows the ratio of high-entropy tokens measured on rollouts generated by the student model during EOPD training, where entropy is computed from the teacher model. High-entropy tokens are defined as token positions whose teacher entropy exceeds a threshold $\tau \, (= 0.8)$, indicating regions where the teacher exhibits higher uncertainty.

As shown in the figure, the proportion of high-entropy tokens is relatively high during the early stages of training and decreases as training progresses. After convergence, the ratio stabilizes at approximately 15% to 20%. This observation motivates the random forward KL setting in §5.8, where forward KL is applied to a randomly selected 20% of token positions. In this setting, the ratio of tokens receiving forward KL is matched to EOPD, while the token selection is independent of the teacher's uncertainty, serving as a controlled baseline.

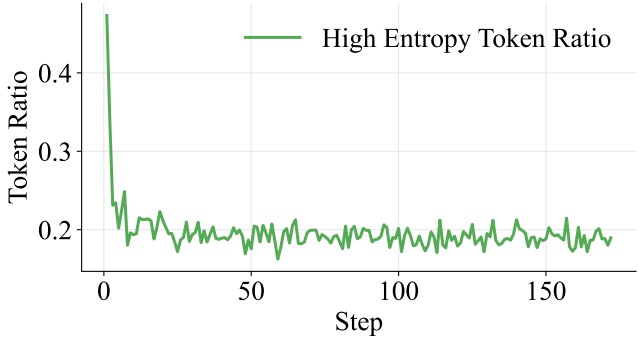

*Figure 9.* Ratio of tokens with high teacher entropy in rollouts produced by the student model during EOPD training.

# F. Tradeoff Between Cumulative Probability Mass and Memory in Top-$k$ Truncation

Forward KL is defined as an expectation over the teacher's full vocabulary distribution, but computing it over the entire vocabulary incurs substantial memory overhead. We therefore approximate the forward KL using only the teacher's top-$k$ tokens in EOPD. To analyze whether a small $k$ preserves sufficient probability mass, we measure the following for different values of $k$: (i) the average cumulative probability mass covered by the top-$k$ tokens; and (ii) the memory required to store the top-$k$ token probabilities and token indices for each batch. Figure 10 shows the tradeoff between cumulative probability mass and memory usage as $k$ varies. While memory increases linearly with $k$, the cumulative probability mass quickly saturates. Empirically, $k = 16$ retains most of the probability mass while incurring a relatively small memory overhead of 144 MiB. Accordingly, we use $k = 16$ in our main experiments.

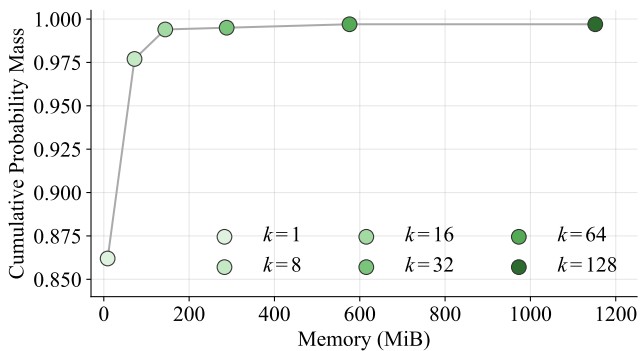

*Figure 10.* Tradeoff between cumulative probability mass and memory usage under top-$k$ truncation. Memory increases linearly with $k$, while the cumulative probability mass quickly saturates.

## G. Results on the Llama Model

*Table 10.* Comparison with other baselines on mathematical reasoning benchmarks using a Llama series model. EOPD achieves competitive performance across benchmarks. **Bold** indicates the best performance and underline indicates second-best.

| Benchmark | Metric | KD | GRPO | OPD | EOPD |
|---|---|---|---|---|---|
| MATH500 | Avg@8 | 42.60 | **45.01** | 43.60 | 44.90 |
| | Pass@8 | 65.80 | 66.00 | 64.80 | **67.20** |
| AMC23 | Avg@8 | 18.43 | 20.31 | 19.06 | **22.18** |
| | Pass@8 | **52.50** | 50.00 | 47.50 | **52.50** |
| AIME24 | Avg@8 | 1.66 | **2.91** | 2.08 | **2.91** |
| | Pass@8 | **13.33** | 10.00 | 10.00 | **13.33** |
| AIME25 | Avg@8 | 1.25 | **2.08** | 1.25 | **2.08** |
| | Pass@8 | 6.67 | **10.00** | 6.67 | **10.00** |

To verify that the proposed EOPD is not limited to a specific model family and can operate effectively on other architectures, we conduct additional experiments on the Llama series (Grattafiori et al., 2024). Specifically, we use Llama-3.2-3B-Instruct as the student model and Llama-3.1-8B-Instruct as the teacher model, and perform training under the same framework. The baseline setup follows §5.1, including KD, GRPO, and OPD. Experimental results reported in Table 10 show that EOPD achieves competitive performance compared to other baselines on the Llama model as well, suggesting that EOPD can be generally applied.

Additionally, we analyze the training dynamics by measuring the forward KL divergence between the teacher and student distributions at token positions where the teacher exhibits high entropy (entropy $\geq 0.8$). As shown in Figure 11, under OPD, the forward KL remains high in these regions and exhibits substantial fluctuations throughout training. This implies that reverse KL–based reward produces unstable signals in regions where the teacher distribution has high entropy, as observed in the toy experiment in §3.2. In contrast, EOPD maintains lower forward KL values in the same high-entropy regions and exhibits more stable training behavior, indicating better preservation of the teacher's distributional structure and uncertainty, which is reflected in the benchmark performance gains.

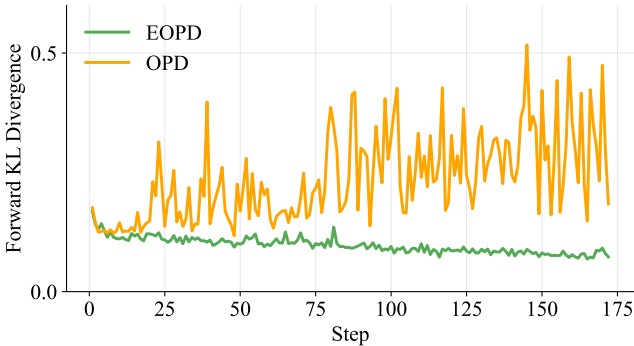

*Figure 11.* Average forward KL divergence measured during training at token positions where the teacher distribution exhibits high entropy (entropy $\geq 0.8$) for the Llama-3.2-3B-Instruct student.

## H. Other Baselines

In this section, we compare EOPD with BD-KD (Amara et al., 2022), AKL (Wu et al., 2025), and ToDi (Jung et al., 2025). These methods adapt the divergence objective based on teacher-student discrepancy measures. Specifically, BD-KD adjusts KL weighting based on entropy difference, AKL uses head-tail distribution mismatch, and ToDi performs token-wise divergence weighting based on vocabulary-level discrepancy. For a fair comparison, we use the official implementations for ToDi and AKL, while BD-KD is implemented following the original formulation. We also align key hyperparameters, including batch size, across all methods.

As shown in Table 11, EOPD achieves superior or comparable performance to existing adaptive KL-based methods across most benchmarks. This suggests that teacher-entropy-based selective forward KL effectively mitigates instability and diversity degradation caused by reverse KL in on-policy distillation. In addition, unlike other baselines, EOPD applies

forward KL only at necessary positions, reducing additional computational overhead.

*Table 11.* Comparison with adaptive KL-based baselines. **Bold** indicates the best performance and underline indicates the second-best performance.

| Benchmark | Metric | BD-KD | AKL | ToDi | EOPD |
|-----------|--------|-------|-----|------|------|
| MATH500 | Avg@8 | 67.45 | 66.43 | 67.43 | **68.73** |
|  | Pass@8 | 86.80 | 85.20 | 86.80 | **87.60** |
| AMC23 | Avg@8 | 38.75 | 37.81 | 38.75 | **41.88** |
|  | Pass@8 | 72.50 | 70.00 | **75.00** | **75.00** |
| AIME24 | Avg@8 | 9.17 | 7.50 | 8.33 | **10.42** |
|  | Pass@8 | 20.00 | 16.67 | 20.00 | **23.33** |
| AIME25 | Avg@8 | 4.58 | 4.58 | **5.83** | **5.83** |
|  | Pass@8 | **16.67** | 13.33 | **16.67** | **16.67** |

## I. Additional Ablations

**Impact of Entropy Threshold $\tau$.** The entropy threshold $\tau$ is a hyperparameter that controls when the forward KL objective is activated based on the teacher model's token-level uncertainty. A lower $\tau$ activates forward KL at more tokens, thereby increasing its overall influence during training, while a higher $\tau$ restricts forward KL to only a small subset of tokens where the teacher exhibits the highest uncertainty. As shown in Table 12, EOPD demonstrates stable performance across a wide range of $\tau$ values, indicating that the method is not highly sensitive to this hyperparameter. Also, we observe an overall trend where Pass@8 performance decreases as $\tau$ increases. This suggests that restricting the application of forward KL can inhibit the transfer of the teacher's uncertainty and diverse reasoning trajectories. Overall, the best performance is obtained at $\tau = 0.8$, and we use this value for all other experiments in this paper.

*Table 12.* Performance variation with respect to the entropy threshold $\tau$ for the Qwen3-1.7B-Base student. EOPD is not highly sensitive to $\tau$, while we observe an overall trend where Pass@8 performance decreases as $\tau$ increases.

| $\tau$ | MATH500 | | AMC23 | |
|--------|---------|--------|-------|--------|
|  | Avg@8 | Pass@8 | Avg@8 | Pass@8 |
| 0.6 | 68.24 | 86.80 | 39.69 | 75.00 |
| 0.8 | 68.73 | 87.60 | 41.88 | 75.00 |
| 1.0 | 67.58 | 84.00 | 41.53 | 72.50 |
| 1.2 | 64.82 | 84.80 | 39.69 | 75.00 |
| 1.4 | 67.61 | 83.80 | 38.72 | 72.50 |
| 1.6 | 68.10 | 84.00 | 39.62 | 70.00 |

**Impact of Forward KL Coefficient $\alpha$.** The forward KL coefficient $\alpha$ controls the influence of the forward KL objective in high-entropy regions. A smaller $\alpha$ may fail to sufficiently transfer the teacher distribution's uncertainty, while an excessively large $\alpha$ can weaken reverse KL's ability to capture the teacher distribution's dominant modes. As shown in Table 13, EOPD demonstrates stable performance across a range of $\alpha$ values, with the best performance achieved around $\alpha = 1.0 \sim 1.2$. This suggests that a balanced combination of forward and reverse KL in high-entropy regions is important for optimal performance. Unless otherwise specified, we use $\alpha = 1.0$ for all experiments in this paper.

*Table 13.* Impact of the forward KL coefficient $\alpha$ for the Qwen3-1.7B-Base student. **Bold** indicates the best performance and underline indicates second-best.

| Benchmark | Metric | $\alpha = 0.6$ | $\alpha = 1.0$ | $\alpha = 1.2$ | $\alpha = 1.5$ |
|---|---|---|---|---|---|
| MATH500 | Avg@8 | 67.37 | 68.73 | **69.55** | 67.50 |
| | Pass@8 | 86.20 | **87.60** | 87.40 | 86.00 |
| AMC23 | Avg@8 | 38.12 | **41.88** | 41.56 | 37.81 |
| | Pass@8 | 72.50 | **75.00** | **75.00** | 67.50 |
| AIME24 | Avg@8 | 8.75 | 10.42 | **10.83** | 9.17 |
| | Pass@8 | 16.67 | **23.33** | **23.33** | **23.33** |
| AIME25 | Avg@8 | 4.58 | **5.83** | 5.42 | 5.00 |
| | Pass@8 | **16.67** | **16.67** | **16.67** | 13.33 |

**Impact of Top-k Selection.** Top-k selection determines the number of teacher tokens used for the forward KL computation. A small $k$ may fail to capture sufficient probability mass from the teacher distribution, while an excessively large $k$ includes low-probability tails and increases computational cost. As shown in Table 14, EOPD shows relatively stable performance across different $k$ values, with the best performance achieved at $k = 16$. This suggests that capturing most of the teacher distribution's probability mass while avoiding unnecessary low-probability tokens is important for effective and efficient training. Unless otherwise specified, we use $k = 16$ for all experiments in this paper.

*Table 14.* Impact of top-k selection for the Qwen3-1.7B-Base student. **Bold** indicates the best performance and underline indicates second-best.

| Benchmark | Metric | $k = 1$ | $k = 8$ | $k = 16$ | $k = 128$ |
|---|---|---|---|---|---|
| MATH500 | Avg@8 | 67.72 | 67.15 | **68.73** | 65.21 |
| | Pass@8 | 84.20 | 85.80 | **87.60** | 87.40 |
| AMC23 | Avg@8 | 38.19 | 38.25 | **41.88** | 39.42 |
| | Pass@8 | 65.00 | 72.50 | **75.00** | **75.00** |
| AIME24 | Avg@8 | 9.17 | **10.42** | **10.42** | 8.75 |
| | Pass@8 | 16.67 | 16.67 | **23.33** | **23.33** |
| AIME25 | Avg@8 | 5.41 | **5.83** | **5.83** | 4.16 |
| | Pass@8 | 13.33 | **16.67** | **16.67** | **16.67** |

