# OpenReview forum: "Entropy-Aware On-Policy Distillation of Language Models"
_ICML.cc/2026/Conference — ICML 2026 regular_

### Official Review · Reviewer_xigx · 2026-02-23

**Soundness:** 2
**Presentation:** 2
**Significance:** 2
**Originality:** 2
**Overall Recommendation:** 3
**Confidence:** 3

**Summary:**

This paper identifies a critical limitation of standard on-policy distillation (OPD) for language models: optimizing solely for Reverse KL (RKL) divergence induces aggressive mode-seeking behavior, which collapses generation diversity and causes training instability in regions where the teacher's distribution exhibits high entropy. To address this, the authors propose Entropy-Aware On-Policy Distillation (EOPD). EOPD introduces a token-level dynamic objective that selectively applies Forward KL (FKL) divergence over the teacher's top-k tokens when the teacher's entropy exceeds a predefined threshold $\tau$, while reverting to the standard clipped RKL objective in low-entropy regions.

**Compliance With Llm Reviewing Policy:**

Affirmed.

**Final Justification:**

I still consider the method to be incremental. However, the authors have addressed most of my concerns, so I will raise my score from 2 to 3. (W2 and Q2 still remain)

**Key Questions For Authors:**

1. Teacher Temperature Ablation: What happens to the performance gap between standard OPD and EOPD if you drastically reduce the teacher model's temperature or top-p threshold? Does the issue of "diversity degradation" disappear when the teacher is forced into a low-entropy state, and could this serve as a simpler, alternative baseline?

2. Computational Cost Trade-off: EOPD requires computing the teacher's entropy and top-k distributions on-the-fly. Could you provide a concrete comparison of the training wall-clock time and memory usage between standard OPD, GRPO, and EOPD? Are the marginal gains in accuracy justified by the extra computational cost?

3. Rigorous GKD Comparison: In Table 6, you compare against "Random" FKL. However, the true strength of GKD [1] lies in tuning its objective weight $\lambda$. Can you provide an ablation study that sweeps the $\lambda$ parameter for GKD to offer a more comprehensive baseline comparison against EOPD?

4. Theoretical Basis for $\tau$: The use of a hard threshold $\tau$ feels somewhat heuristic. Did you experiment with a continuous, differentiable weighting scheme (e.g., dynamically interpolating between RKL and FKL based on a continuous function of the teacher's entropy)? If so, how did it perform, and why was the hard threshold chosen instead?FKL Isolation: Did you attempt an ablation where you only apply FKL to the high-entropy tokens (and apply no loss/zero gradients to low-entropy tokens)? This would help isolate whether the performance gain comes strictly from matching high-entropy states or from the synergistic combination of RKL and FKL.

**Limitations:**

No. While the authors include a brief "Impact Statement" discussing general societal impacts, they lack a dedicated discussion of the methodological limitations of EOPD.

Suggestion on "limitations":
- Dependence on Expert Teacher Model
- Hyperparameter Tuning: Although an ablation study is provided for the threshold $\tau$, the method still introduces new hyperparameters (like $\tau$ and the $k$ size for top-k selection)  that may require empirical tuning when adapting the framework to completely new domains or different teacher-student architecture pairs.

**Strengths And Weaknesses:**

**Strengths:**

- Insightful Entropy Perspective: The diagnosis of OPD's failures through the lens of token-level entropy is a highly intuitive and physically meaningful contribution. In reasoning tasks, high-entropy tokens are not merely noise; they often encode multiple valid reasoning paths and critical decision nodes. The empirical verification of diversity collapse under RKL provides a strong motivation for the proposed method.

- Comprehensive Multi-Scale Validation: The authors rigorously tested their hypothesis across three different student model capacities (0.6B, 1.7B, and 4B). This robust scaling analysis effectively demonstrates that the benefits of EOPD are consistent and not artifacts of a specific model size.

**Weaknesses:**

- Missing Ablation on Teacher Distribution Parameters: The core premise of EOPD relies heavily on the natural occurrence of high-entropy states in the teacher. However, the authors do not investigate whether artificially sharpening the teacher's distribution (e.g., drastically reducing temperature or lowering top-p during teacher generation/querying) would mitigate the RKL instability entirely. If a low-temperature teacher inherently solves the diversity collapse issue, the necessity of the EOPD objective might be undermined.

- Marginal Empirical Gains vs. Added Computational Cost: While the method shows improvement, the gains are not universally significant. In Table 2, approximately 1/3 of the reported metrics show EOPD either tying with or slightly underperforming baseline methods like standard OPD or GRPO. Furthermore, computing the teacher's token-level entropy and querying top-k probabilities for the FKL term inherently increases the computational overhead compared to standard OPD. The paper completely lacks an analysis of training FLOPs, memory footprint, or wall-clock time to justify this trade-off.

- Lack of Theoretical Justification for the Hard Threshold: The method employs a hard, discrete threshold $\tau$ (e.g., $\tau = 0.8$) to switch between RKL and FKL. This step-function approach lacks a rigorous theoretical foundation. A continuous transition mechanism (such as an entropy-weighted interpolation between RKL and FKL) would arguably be more mathematically sound and potentially more stable.

- Incomplete Comparison with Generalized KD (GKD): The paper briefly mentions a variant of GKD [1] in Footnote 3 and tests a "Random FKL" baseline. However, GKD fundamentally relies on interpolating distributions or divergences using a mixing coefficient $\lambda$:

$L\_{\text{GKD}}(\theta) := (1 - \lambda) \mathbb{E}\_{(x,y) \sim (X,Y)} \left[ \mathcal{D}(p\_{\text{T}} \| p\_{\text{S}}^{\theta})(y|x) \right] + \lambda \mathbb{E}\_{x \sim X} \left[ \mathbb{E}\_{y \sim p\_{\text{S}}(\cdot|x)} \left[ \mathcal{D}(p\_{\text{T}} \| p\_{\text{S}}^{\theta})(y|x) \right] \right]$

The authors need to provide a direct comparison against this formalized GKD objective, including detailed hyperparameter tuning for $\lambda$, to strictly prove that their dynamic, entropy-aware routing is superior to a well-tuned static interpolation.

- Unexplored Gradient Mechanics: The paper defines the per-token advantage using log-probability differences but does not explore the gradient mechanics regarding the stop-gradient operator on this advantage term. Investigating whether backpropagating through the probabilities alters exploration dynamics would add technical depth.

---

> ### Author Rebuttal · Authors · 2026-03-31
>
> Dear reviewer xigx,
>
> We appreciate your reviewing efforts and meaningful suggestions, which significantly contributed to improving our work. Below, we address each of your concerns.
>
> Due to space limits, we report MATH and AMC in the main text, with additional results provided via an anonymized link.
>
> **[W1, Q1] Teacher distribution**
> We conducted additional experiments with standard OPD using lower teacher temperatures (0.5 and 0.7), and EOPD outperformed both settings.
> This is consistent with our motivation: enabling stable transfer of the teacher’s diversity. Lowering the teacher temperature reduces diversity, making it fundamentally different from our approach.
>
> Figure: [https://i.imgur.com/ynniWwO.png]
>
> | Benchmark | Metric | OPD-T0.5 | OPD-T0.7 | EOPD |
> |-|-|-|-|-|
> | Math500  | Avg@8  | 66.20 | 67.45 | 68.73 |
> |          | Pass@8 | 83.80 | 85.20 | 87.60 |
> | AMC23    | Avg@8  | 37.19 | 39.38 | 41.88 |
> |          | Pass@8 | 65.00 | 67.50 | 75.00 |
>
> [https://i.imgur.com/tz8y1wo.png]
>
> **[W2, Q2] Marginal gain and Cost**
>
> For the cost analysis, please refer to our response to reviewer phsP, [Q3].
> Overall, EOPD adds limited overhead over OPD, while improving uncertainty transfer in high-entropy regions and enhancing Pass@k performance.
>
> Also, our key contribution is improved transfer of the teacher’s uncertainty in high-entropy regions. Empirically, EOPD outperforms OPD on average, with larger gains on Pass@8, supported by entropy and forward-KL analyses.
>
> **[W3, Q4] Justification for the Hard Threshold, FKL isolation**
> Our main point is not the specific choice of hard threshold, but the need for an entropy-aware approach: reverse KL is effective in low-entropy regions, while forward KL better captures multiple plausible continuations in high-entropy regions.
>
> Following the reviewer’s suggestion, we tested a continuous weighting $\alpha \log(1+H)$. With $\alpha=1.0$, performance was comparable to hard-threshold EOPD, while $\alpha=1.5$ was slightly worse (see the table below). This suggests that the benefit comes from selective application in high-uncertainty regions, rather than softness itself.
>
> The hard-threshold design also has a practical advantage: it is more memory-efficient (please refer to our response to reviewer phsP, [Q3]) than applying forward KL across all positions.
>
> | Benchmark | Metric | soft (alpha=1.0) | soft (alpha=1.5) | EOPD |
> |-|-|-|-|-|
> | Math500  | Avg@8  | 68.48 | 67.43 | 68.73 |
> |          | Pass@8 | 87.40 | 85.00 | 87.60 |
> | AMC23    | Avg@8  | 41.76 | 38.75 | 41.88 |
> |          | Pass@8 | 72.50 | 72.50 | 75.00 |
>
> [https://i.imgur.com/CGCnEWC.png]
>
>
> **FKL Isolation**
> Following the reviewer’s suggestion, we conducted an ablation in which forward KL is applied only to high-entropy teacher tokens, with no distillation loss on low-entropy tokens (FKL isolation).
> This setting consistently underperforms EOPD across all benchmarks, indicating that forward KL alone is insufficient and that reverse KL remains important for capturing dominant modes.
>
> | Benchmark | Metric | FKL Isolation | EOPD |
> |-|-|-|-|
> | Math500  | Avg@8  | 64.83 | 68.73 |
> |          | Pass@8 | 85.80 | 87.60 |
> | AMC23    | Avg@8  | 35.63 | 41.88 |
> |          | Pass@8 | 65.00 | 75.00 |
>
> [https://i.imgur.com/2mBJCBK.png]
>
>
> **[W4, Q3] Comparison with GKD**
>
> To address this concern, we compare against a GKD baseline [1]. Following GKD, we vary the fraction of on-policy student-generated outputs $\lambda$ and optimize divergence on both the fixed dataset $(X, Y)$ and student outputs.
>
> As GKD suggests forward KL for math reasoning, we adopt forward KL and sweep $\lambda \in {0.5, 0.8, 1.0}$. The dataset $(X, Y)$ is generated by the same teacher to align supervision with its distribution.
>
> EOPD still outperforms this tuned baseline at $\lambda = 1.0$, indicating that the gains come from entropy-aware routing rather than static interpolation. We will clarify this in the revised version.
>
> | Benchmark | Metric | GKD (lambda=0.5) | GKD (lambda=0.8) | GKD (lambda=1.0) | EOPD |
> |-|-|-|-|-|-|
> | Math500  | Avg@8  | 64.53 | 65.28 | 67.20 | 68.73 |
> |          | Pass@8 | 84.40 | 84.20 | 85.00 | 87.60 |
> | AMC23    | Avg@8  | 35.62 | 37.50 | 38.44 | 41.88 |
> |          | Pass@8 | 67.50 | 70.00 | 67.50 | 75.00 |
>
> [https://i.imgur.com/ocuAc9Q.png]
>
> [1] Agarwal, Rishabh, et al. "On-policy distillation of language models: Learning from self-generated mistakes."
>
> **[W5] Gradient mechanics**
> Thank you for the suggestion. In our implementation, we apply a stop-gradient to the advantage, treating it as a detached scalar weight following the standard policy-gradient formulation. Allowing gradients through the advantage would introduce additional terms and deviate from this objective.
> However, we agree that analyzing backpropagation through the advantage term and its effect on exploration dynamics is interesting; we will briefly discuss this mechanism in the revision.
>
>
> [Limitations]: Please check our response to reviewer LUi6, [W2, Q2]

---

> > ### Author Rebuttal · Reviewer_xigx · 2026-04-03
> >
> > I still consider the method to be incremental. However, the authors have addressed most of my concerns, so I will raise my score from 2 to 3. (W2 and Q2 still remain)

---

> > > ### Author Response · Authors · 2026-04-06
> > >
> > > Thank you for the update and for raising your score. We appreciate that most of your concerns have been addressed.
> > >
> > > Additionally, we have provided further analysis clarifying how our method **differs from prior work** and **demonstrating the empirical gains** of our approach. We believe this can address the remaining concerns.
> > >
> > > In particular, we provide a detailed comparison with related baselines highlighting the difference in the switching criterion and its impact on performance. We kindly refer the reviewer to our additional response to **Reviewer LUi6** for more details.
> > >
> > > Again, thank you for the valuable feedback.
> > >
> > > Many thanks, Authors

---

### Official Review · Reviewer_phsP · 2026-03-10

**Soundness:** 3
**Presentation:** 3
**Significance:** 3
**Originality:** 3
**Overall Recommendation:** 4
**Confidence:** 3

**Summary:**

This paper studies a limitation of reverse-KL-based on-policy distillation for language models, namely its tendency to become mode-seeking and to under-transfer teacher uncertainty in high-entropy regions. To address this, the paper proposes Entropy-Aware On-Policy Distillation (EOPD), which augments the standard reverse-KL objective with a forward-KL term selectively activated when the teacher’s token-level entropy exceeds a threshold. The paper supports the motivation through token-entropy analysis and a toy instability study, and reports gains on several math reasoning benchmarks, with additional out-of-domain and entropy-focused analyses.

**Compliance With Llm Reviewing Policy:**

Affirmed.

**Final Justification:**

All my concerns are addressed

**Key Questions For Authors:**

- How does the proposed method compare empirically with ToDi or other token-wise divergence adaptation methods that also combine forward and reverse KL? Would the entropy-based switching still provide advantages over probability-gap–based criteria?

- The experiments focus primarily on Qwen3 models and math reasoning benchmarks. Would the same improvements hold for other model families or non-reasoning tasks where the structure of teacher uncertainty may differ?

- What is the practical computational overhead of EOPD compared to standard OPD in terms of training time, teacher queries, and memory usage?

**Limitations:**

yes

**Strengths And Weaknesses:**

**strengths:**

- Clear and well-motivated problem formulation.

- Simple and practically appealing method with consistent empirical gains on reasoning tasks.

**Weaknesses:**

- Most experiments are concentrated on Qwen3 teacher-student settings and primarily on mathematical reasoning benchmarks. Although the out-of-domain results are a useful addition, the empirical scope is limited in my opinion.

- Incomplete baseline set. For example ToDi (Token-wise Distillation) also adapts the divergence direction at the token level by combining forward and reverse KL based on the teacher–student probability mismatch. Because the proposed method also performs token-wise divergence selection, ToDi represents a particularly relevant baseline. The absence of an empirical comparison makes it difficult to determine whether the gains come from the entropy-based criterion specifically or from the general idea of adaptive KL switching.

---

> ### Author Rebuttal · Authors · 2026-03-31
>
> Dear reviewer phsP,
> We appreciate your efforts in reviewing the manuscript and providing meaningful suggestions. Below, we address each concern in detail.
>
> Due to space limits, we report MATH and AMC in the main text, with additional results provided via an anonymized link.
>
>
> **[W1, Q2] Generalization to other model, domain, benchmark**
>
> **Different model families**
>
> We thank the reviewer for this suggestion. To assess generalization across model families, we conducted additional experiments using a LLaMA-based teacher, comparing EOPD against OPD-FKL (Forward KL under OPD framework) and OPD-RKL (Reverse KL under OPD framework).
>
> When using the LLaMA-based teacher, EOPD consistently outperforms both OPD-FKL and OPD-RKL. This indicates that EOPD's entropy-aware selection strategy remains effective across different model families.
>
> | Benchmark | Metric | OPD-RKL | OPD-FKL | EOPD |
> |-|-|-|-|-|
> | Math500  | Avg@8  | 43.60 | 43.77 | 44.90 |
> |          | Pass@8 | 64.80 | 66.20 | 67.20 |
> | AMC23    | Avg@8  | 19.06 | 20.63 | 22.18 |
> |          | Pass@8 | 47.50 | 47.50 | 52.50 |
>
> [https://i.imgur.com/s5Zj0Hi.png]
>
> **Different domain**
>
> To evaluate generalization, we train on an additional dataset (Alpaca, randomly selecting 10k prompts) and compare EOPD with OPD-FKL and OPD-RKL. The models are evaluated on AlpacaEval 2 and an additional benchmark, IFEval.
> The table shows that EOPD outperforms baselines beyond the math domain. We will include these results in the revised manuscript.
> | Benchmark | OPD-FKL | OPD-RKL | EOPD |
> |-|-|-|-|
> | AlpacaEval | 25.15 ± 1.55 | 24.77 ± 1.56 | 26.33 ± 1.53 |
> | IFEval (prompt_level_strict) | 30.12 ± 2.13 | 32.72 ± 2.02 | 33.46 ± 2.03 |
>
>
> **[W2, Q1] How EOPD differs from prior approaches**
>
> EOPD differs from prior adaptive KL methods by **not focusing on discrepancy, but instead addressing reverse KL instability and diversity reduction**. These issues become particularly critical in on-policy distillation settings, and we mitigate them via teacher-uncertainty-guided forward KL.
>
> Existing approaches such as BD-KD, AKL, and ToDi adapt the KL objective based on different forms of teacher–student discrepancy, including entropy gap, head/tail distribution mismatch, and vocabulary index-level differences, respectively. These methods share a common goal of improving distribution alignment by more precisely estimating discrepancy.
>
> However, our approach is motivated by a different perspective. We focus on the behavior of reverse KL in on-policy distillation, where it can lead to instability and reduced diversity, particularly under high-entropy teacher distributions. EOPD addresses this by selectively incorporating forward KL in such regions based on teacher uncertainty, directly targeting this problem.
>
> As suggested by the reviewer, we compare our method with ToDi, one of the most directly comparable baselines among token-wise divergence adaptation methods.
>
> We use the official ToDi implementation and align key hyperparameters (e.g., batch size) for a fair comparison. EOPD outperforms ToDi (see the table below), suggesting that the gains stem from the switching criterion rather than adaptive switching alone.
>
> ToDi also incurs higher computational cost, requiring the full teacher distribution and per-token vocabulary-wise weighting. In contrast, EOPD avoids this overhead and is more efficient, achieving 47.8 sec/step vs. 71.2 sec/step for ToDi.
>
> | Benchmark | Metric | ToDi | EOPD |
> |-|-|-|-|
> | Math500  | Avg@8  | 67.43 | 68.73 |
> |          | Pass@8 | 86.80 | 87.60 |
> | AMC23    | Avg@8  | 38.75 | 41.88 |
> |          | Pass@8 | 75.00 | 75.00 |
>
> [https://i.imgur.com/tCljFdI.png]
>
>
>
> **[Q3] Computational Overhead**
>
> To address the reviewer’s concern about computational overhead, we analyze training time, teacher queries, and memory usage on a 4×A100 80GB GPU setup. Overall, the additional cost of EOPD relative to OPD is limited. In addition, while EOPD uses a single rollout per prompt as in OPD, GRPO requires eight rollouts, making it substantially more expensive. We support this claim with detailed measurements below.
>
> **Teacher query**
>
> EOPD requires no additional teacher forward passes compared to OPD.  In both methods, the teacher is queried once per student-generated token position.
>
> **Training time**
>
> EOPD adds a small overhead for forward KL computation (top-k extraction, renormalization, and loss calculation), measured as an additional 2.16 seconds per training step (8 microbatches per GPU), which is about 4.5% of the average 47.8-second step time. This overhead is marginal, as most time is spent on the student’s on-policy generation (~37.7 seconds).
>
> **Memory usage**
> For forward KL, tensors for selected tokens are stored per microbatch, incurring an average of 219.3 MB when applied selectively, which is manageable.
> In contrast, applying forward KL to all tokens requires an average of 1617.5 MB, showing a clear memory advantage (7.37× reduction) of selective application.

---

> > ### Author Rebuttal · Reviewer_phsP · 2026-04-04
> >
> > I thank the authors for the detailed response. While the method is simple (and to some extent incremental), the additional empirical results show improvement compared to baselines. Hence, I am more confident in my original assessment (weak accept).

---

> > > ### Author Response · Authors · 2026-04-06
> > >
> > > Thank you for your response.
> > > We are happy to hear that we have addressed your concerns. Following your suggestion, we will further clarify the novelty of our method in the final manuscript.
> > > If you have any further questions or suggestions, please do not hesitate to let us know. Again, thank you for the valuable suggestion and your positive assessment of our work.
> > >
> > > Many thanks,
> > > Authors

---

### Official Review · Reviewer_LUi6 · 2026-03-12

**Soundness:** 2
**Presentation:** 3
**Significance:** 2
**Originality:** 2
**Overall Recommendation:** 4
**Confidence:** 4

**Summary:**

This paper studies the limitations of OPD. In the standard OPD framework, the student model matches the teacher's token-level distribution via reverse KL along its own generated trajectories. The paper identifies that the mode-seeking property of reverse KL causes diversity degradation and unstable learning signals at positions where the teacher distribution has high entropy. To address this, the authors propose Entropy-Aware On-Policy Distillation, which augments the standard OPD reverse KL objective with a forward KL term when the teacher's entropy exceeds a threshold. The forward KL is computed only over the teacher's top-k tokens to control computational overhead. Experiments are conducted on three scales of Qwen3 student models (0.6B, 1.7B, 4B) with Qwen3-8B (non-thinking mode) as the teacher, yielding significant improvements across multiple math benchmarks.

**Compliance With Llm Reviewing Policy:**

Affirmed.

**Final Justification:**

Some of my concerns are resolved. Although the claims regarding novelty still fail to articulate the fundamental differences from related works, authors have tried their best to compare EODP with them. I would raise my score.

**Key Questions For Authors:**

1. In Table 6, full forward KL performs very close to EOPD. Does this suggest that the core mechanism of "entropy-based position selection" has limited contribution to performance improvement, and that the advantage of EOPD over the FULL setting may primarily lie in efficiency and cost?
2. Have the coefficient of the forward KL term and the choice of top-k been tuned? How would performance change if the coefficient were set to a value other than 1, or if the hard threshold switching were replaced with a continuous weighting function?
3. Could the authors supplement a baseline of pure forward KL distillation under the on-policy framework?

**Limitations:**

yes

**Strengths And Weaknesses:**

Strengths:
1. On-policy distillation is an important paradigm for efficiently training reasoning models, and analyzing the behavior of reverse KL in high-entropy regions has practical value.
2. The paper conducts experiments across three model scales and evaluates on six mathematical benchmarks and three OOD benchmarks. The experimental results and analysis effectively demonstrate the practical gains brought by EOPD's preservation of model diversity.
3. Through comparisons with entropy bonus and advantage shaping, combined with training dynamics analysis, the authors clearly demonstrate that high entropy does not indicate alignment with the teacher. This finding provides valuable insight for understanding exploration strategies.
4. EOPD only requires adding an entropy-based forward KL term to the existing on-policy distillation pipeline, making it easy to reproduce and deploy in practice.

Weaknesses:
1. The core contribution of this paper is the dynamic combination of forward KL and reverse KL based on entropy signals, but similar ideas have already been proposed in multiple related works. BD-KD [1] proposes a sample-level FKL/RKL switching mechanism based on teacher-student entropy difference. AKL [2] dynamically assigns FKL/RKL weights and theoretically argues that mode-seeking/mean-seeking are merely different optimization paths that ultimately converge. Although the paper mentions related work in the related work section, it does not sufficiently discuss how EOPD differs from these works at the methodological level, thereby weakening the claimed contribution and novelty.
2. Several critical aspects lack ablation. EOPD uses a hard indicator to switch between forward KL and fixes the weight of the forward KL term at 1, yet the weighting between forward and reverse KL can significantly impact performance. Furthermore, a natural alternative is to use a continuous weighting function that allows the influence of forward KL to vary smoothly with entropy, but the paper does not conduct ablation experiments on weight settings. Finally, the forward KL is computed only over the teacher's top-k tokens, yet the paper does not discuss the impact of smaller or larger k on the forward KL and final results.
3. The toy experiment removes various dependencies, and these simplifications make the transferability of the experimental findings to real LLM training questionable.
4. All experiments use only a single teacher model. Different teachers may have substantially different entropy distribution characteristics, and whether EOPD generalizes to other model families is unknown. Moreover, the paper lacks a critical baseline: pure forward KL distillation under the on-policy framework, making it hard to disentangle the contributions of "adding forward KL" versus "entropy-based position selection."

[1] BD-KD: Balancing the Divergences for Online Knowledge Distillation, 2024

[2] Rethinking Kullback-Leibler Divergence in Knowledge Distillation for Large Language Models, 2024

---

> ### Author Rebuttal · Authors · 2026-03-31
>
> We appreciate your efforts in reviewing the manuscript and providing meaningful suggestions. Below, we address each of your concerns.
>
> Due to space limits, we report MATH and AMC in the main text, with additional results provided via an anonymized link.
>
> **[W1] How EOPD differs from prior approaches**
>
> Please check the response to reviewer phsP, [W2, Q1].
>
>
> **[W2, Q2] Additional ablation studies**
>
> - Weighting between forward and reverse KL
>
> To examine the effect of the forward KL coefficient, we conduct experiments by varying it over 0.6, 1.0, 1.2, and 1.5. Performance peaks at coefficients 1.0–1.2, while both lower (0.6) and higher (1.5) values reduce performance. This suggests insufficient forward KL is ineffective, whereas excessive forward KL weakens reverse KL’s ability to capture the teacher distribution’s dominant modes. Overall, a balanced combination of forward and reverse KL in high-entropy regions is crucial for optimal performance.
>
> | Benchmark | Metric | w = 0.6 | w = 1.0 | w = 1.2 | w = 1.5 |
> |-|-|-|-|-|-|
> | Math500  | Avg@8  | 67.37   | 68.73   | 69.55   | 67.50   |
> |          | Pass@8 | 86.20   | 87.60   | 87.40   | 86.00   |
> | AMC23    | Avg@8  | 38.12   | 41.88   | 41.56   | 37.81   |
> |          | Pass@8 | 72.50   | 75.00   | 75.00   | 67.50   |
>
> [https://i.imgur.com/uPa76SQ.png]
>
> - Continuous weighting function that allows the influence of forward KL to vary smoothly with entropy
>
> Please check the response to reviewer xigx, [W3, Q4]
>
> **Impact of smaller or larger k on the forward KL**
> To analyze the effect of the top-$k$ used in the forward KL computation, we evaluate $k \in {1, 8, 16, 128}$.
>
> | Benchmark | Metric | k = 1 | k = 8 | k = 16 | k = 128 |
> |-|-|-|-|-|-|
> | Math500  | Avg@8  | 67.72 | 67.15 | 68.73 | 65.21 |
> |          | Pass@8 | 84.20 | 85.80 | 87.60 | 87.40 |
> | AMC23    | Avg@8  | 38.19 | 38.25 | 41.88 | 39.42 |
> |          | Pass@8 | 65.00 | 72.50 | 75.00 | 75.00 |
>
> [https://i.imgur.com/XA1Ua2M.png]
>
> When $k$ is small (e.g., $k=1$), the teacher distribution is inadequately captured, whereas large $k$ (e.g., $k=128$) includes low-probability tails, reducing efficiency and degrading performance. Moderate values (e.g., $k=16$) provide stable performance by covering most of the probability mass, while avoiding excessive computational and memory costs.
>
>
> **[W3] Toy experiments**
>
> The toy experiment is intentionally simplified: its purpose is not to replicate real LLM training, but to provide a controlled, interpretable setting for isolating why reverse KL can produce unstable learning signals in high-entropy regions of the teacher distribution.
>
> To examine more realistic settings, we analyze optimization dynamics in LLM training. Tracking gradient norms in the LLaMA family, EOPD shows more stable behavior than reverse KL-based OPD. This observation aligns with findings from the toy setting. We will include this analysis in the revised manuscript.
>
> [https://i.imgur.com/BdKrUe0.png]
>
>
> **[W4] Generalization to other model**
>
> Please check the response to reviewer phsP, [W1, Q2].
>
> **[W4, Q3] Pure forward KL distillation under the on-policy framework**
>
> We thank the reviewer for this important suggestion. To disentangle the contributions of forward KL versus entropy-based position selection, we evaluated a pure forward KL distillation baseline under the on-policy framework (OPD-FKL). The results are shown below
> | Benchmark | Metric | OPD-FKL | OPD-RKL (Current OPD) | EOPD |
> |-|-|-|-|-|
> | Math500  | Avg@8  | 67.20 | 67.76 | 68.73 |
> |          | Pass@8 | 85.00 | 84.80 | 87.60 |
> | AMC23    | Avg@8  | 38.44 | 39.06 | 41.88 |
> |          | Pass@8 | 67.50 | 70.00 | 75.00 |
>
> OPD-FKL underperforms OPD-RKL, while EOPD surpasses both, indicating that EOPD’s gains are not due to forward KL alone. Instead, they highlight the entropy-aware selection mechanism, where forward KL is applied selectively to high-uncertainty positions rather than uniformly.
>
> [https://i.imgur.com/3psJY44.png]
>
>
> **[Q1] Advantage of EOPD over the FULL setting**
>
> We respectfully disagree about this point. Avg@8 shows a clear improvement of EOPD over FULL at 1.7B, and the gap widens at 4B (see the table below), suggesting better scaling behavior.
> The reason why selective application outperforms uniform application aligns with our intuition: when a capacity gap exists, uniformly applying forward KL forces the student to match low-probability tails even at low-entropy positions. EOPD instead applies reverse KL at low-entropy positions and forward KL at high-entropy ones, enabling more targeted transfer.
> EOPD also provides an additional efficiency gain. Under $\tau=0.8$, selectively applying forward KL reduces memory usage by 7.37× (please refer to our response to reviewer phsP, [Q3]).
>
> | Benchmark | Metric | Full FKL | EOPD |
> |-|-|-|-|
> | Math500  | Avg@8  | 78.82 | 80.20 |
> |          | Pass@8 | 90.20 | 93.00 |
> | AMC23    | Avg@8  | 58.40 | 60.94 |
> |          | Pass@8 | 82.50 | 87.50 |
>
> [https://i.imgur.com/fPgr96w.png]

---

> > ### Author Rebuttal · Reviewer_LUi6 · 2026-04-04
> >
> > Thank you for the additional experiments, which strengthen the empirical picture. However, my core novelty concern remains: EOPD's KL switching is structurally very similar to BD-KD/AKL/... The rebuttal compares only against ToDi, which cannot isolate whether the entropy-based criterion itself is the source of improvement. I maintain my current score, but would be willing to raise it in subsequent discussions if a much more comprehensive comparison with related work or a detailed discussion of differences is provided.

---

> > > ### Author Response · Authors · 2026-04-06
> > >
> > > Thank you for the insightful comment. We agree that comparing only with ToDi is insufficient to isolate the effect of the switching criterion.
> > >
> > > Accordingly, we conducted additional comparisons with BD-KD and AKL. Since BD-KD is a method originally proposed for image classification, we implemented it on our code by following the original formulation that adjusts KL weighting based on the entropy difference between teacher and student, while AKL was implemented using the official repository. We also aligned key hyperparameters such as batch size, and kept method-specific hyperparameters consistent with the original papers.
> > >
> > > EOPD shows consistent improvements over both methods (see Table below). In addition, analyzing forward KL on high-entropy teacher tokens, EOPD achieves lower values, indicating better alignment with the teacher distribution in these regions.
> > > Figure: [https://i.imgur.com/qJC5n0E.png]
> > >
> > > This difference arises from the difference in the criteria used by each method.
> > > BD-KD adjusts the KL based on the entropy difference between teacher and student. However, **even when the teacher and student have similar entropy, their actual token distributions can differ**, and in such cases BD-KD may fail to adequately reflect regions where the teacher has high uncertainty.
> > > AKL also relies on a criterion based on **head–tail distribution differences** that is not directly related to teacher uncertainty, and therefore does not clearly capture such regions.
> > > In contrast, EOPD uses the teacher’s entropy itself as the criterion, allowing it to **directly identify regions where the teacher has high entropy** and apply forward KL at those positions.
> > >
> > > Therefore, the improvement comes from the difference in the criterion for applying forward KL, which addresses the issue that reverse KL becomes unstable and reduces diversity in high-entropy teacher regions in on-policy distillation. Furthermore, while BD-KD and AKL apply adaptive KL to all tokens, EOPD selectively applies forward KL only to necessary tokens, reducing unnecessary computation, resulting in a training time of 47.8s compared to BD-KD (58.6s) and AKL (61.3s), demonstrating better efficiency.
> > >
> > > We will reflect these results in the revised manuscript.
> > >
> > >
> > > |Benchmark|Metric|BD-KD|AKL|EOPD|
> > > |-|-|-|-|-|
> > > |Math500|Avg@8|67.45|66.43|68.73|
> > > ||Pass@8|86.80|85.20|87.60|
> > > |AMC23|Avg@8|38.75|37.81|41.88|
> > > ||Pass@8|72.50|70.00|75.00|
> > > |AIME24|Avg@8|9.17|7.50|10.42|
> > > ||Pass@8|20.00|16.67|23.33|
> > > |AIME25|Avg@8|4.58|4.58|5.83|
> > > ||Pass@8|16.67|13.33|16.67|
> > > |GPQA|Avg@8|28.34|28.50|31.50|
> > > ||Pass@8|76.36|74.74|81.31|

---

### Decision · Program_Chairs · 2026-04-30

**Decision:**

Accept (regular)

**Comment:**

Reviewers agreed on the problem motivation: reverse-KL-based on-policy distillation can be brittle in high-entropy teacher regions, and a simple practical fix would be useful. They also agreed that the paper is clearly written and that the proposed method is easy to integrate into existing distillation pipelines. The rebuttal added meaningful empirical support, including additional comparisons, broader model/task coverage, and efficiency discussion, which improved the paper's empirical case.

The main remaining disagreement concerned originality. Across the reviews and discussion, the central question was whether the entropy-based switching rule is sufficiently distinct from prior adaptive-KL or mixed-KL distillation methods, or whether the current method should be viewed as an incremental but useful variation. One reviewer stayed positive and became more confident after the rebuttal, another improved from reject to weak reject, and one reviewer remained concerned that the contribution is still too incremental. After reading the discussion, I think the paper is technically sound and practically useful, and the rebuttal substantially reduced soundness-related concerns. At the same time, the remaining novelty concerns make this a borderline case. On balance, I lean toward weak accept rather than reject because the empirical support is stronger after rebuttal and the remaining debate is primarily about incrementality rather than correctness.